# Structure and mechanism of DNA delivery of a gene transfer agent

Pavol Bárdy [1], Tibor Füzik [2], Dominik Hrebík[2], Roman Pantůček [1], J. Thomas Beatty [3] & Pavel Plevka [2 ✉]

Alphaproteobacteria, which are the most abundant microorganisms of temperate oceans, produce phage-like particles called gene transfer agents (GTAs) that mediate lateral gene exchange. However, the mechanism by which GTAs deliver DNA into cells is unknown. Here we present the structure of the GTA of *Rhodobacter capsulatus* (RcGTA) and describe the conformational changes required for its DNA ejection. The structure of RcGTA resembles that of a tailed phage, but it has an oblate head shortened in the direction of the tail axis, which limits its packaging capacity to less than 4,500 base pairs of linear double-stranded DNA. The tail channel of RcGTA contains a trimer of proteins that possess features of both tape measure proteins of long-tailed phages from the family *Siphoviridae* and tail needle proteins of short-tailed phages from the family *Podoviridae*. The opening of a constriction within the RcGTA baseplate enables the ejection of DNA into bacterial periplasm.

[1] Department of Experimental Biology, Faculty of Science, Masaryk University, 625 00 Brno, Czech Republic. [2] Central European Institute of Technology, Masaryk University, 625 00 Brno, Czech Republic. [3] Department of Microbiology and Immunology, University of British Columbia, Vancouver V6T 1Z3 BC, Canada. ✉email: pavel.plevka@ceitec.muni.cz

Gene transfer agents (GTAs) are DNA-containing phage-like particles produced by some species of bacteria and archaea[1]. GTAs are common in alphaproteobacteria from the order Rhodobacterales, which are the most abundant species in temperate oceans[2,3]. The genes encoding proteins that form *Rhodobacter capsulatus* GTA particles (RcGTA) are derived from phage DNA, which has been integrated into the bacterial genome[4]. However, the phage genes for the regulation of protein expression and DNA replication are absent, and cellular pathways control the synthesis of RcGTA proteins[5]. The production of the RcGTA is stimulated by nutrient depletion, which induces entry of *R. capsulatus* into the stationary phase, and high population density detected by quorum sensing[6–8]. Similarly, the recipient capability of *R. capsulatus* cells for RcGTA as well as competence systems of naturally transformable bacteria are highest in the stationary phase, which can be induced by limited availability of nutrients[7,9,10]. The initiation of production of RcGTA is a stochastic process, which limits the expression of RcGTA genes to less than 1% of cells in the population[11,12]. Phage-derived holins and endolysins enable disruption of RcGTA-producing cells to release the particles[11].

RcGTA particles resemble tailed phages from the family *Siphoviridae*[13]. Each RcGTA packages a 4.0–4.5 kilobase-long fragment of double-stranded DNA[14]. RcGTA particles encapsulate all genes of *R. capsulatus*; however, the genes encoding the proteins forming the RcGTA particle are packaged with a lower frequency than other regions of the bacterial genome[14]. It was speculated that high levels of RcGTA gene transcription in the RcGTA-producing cells cause a reduction in their packaging frequency[14]. Unlike most phages, RcGTA delivers DNA into the periplasm between the outer and cytoplasmic membranes of the recipient Gram-negative cells of *Rhodobacter*[15]. It has been shown that translocation of the DNA to the cytoplasm requires competence-derived systems for the uptake of DNA from the periplasm[15]. The internalized DNA may be incorporated into the genome of the recipient bacteria by homologous recombination[15,16]. Lateral gene transfer mediated by GTAs has been demonstrated in several bacterial species, and homologues of RcGTA genes have been found in thousands of bacterial genomes[17]. The increased frequency of genetic exchange due to GTAs may affect bacterial adaptation and evolution[18]. Furthermore, RcGTA has been used as a tool for the genetic manipulation of bacteria[19]. However, the structure and molecular mechanisms of DNA delivery by RcGTA are unknown.

Here, we use cryo-electron microscopy (cryo-EM) to determine the structures of RcGTA before and after DNA ejection, and to visualize the interactions of RcGTA with *R. capsulatus* cells. The asymmetric reconstruction of the native particle of RcGTA has an overall resolution of 4.3 Å, and symmetrized reconstructions of the head, portal complex, tail, and baseplate have resolutions of 3.3–4.5 Å.

## Results and discussion

### An oblate capsid limits DNA packaging capacity of RcGTA.
Particles of RcGTA have heads with a diameter of 38-nm and 49-nm long tails (Fig. 1a, b, Supplementary Figs. 1–3, Supplementary Tables 1–3). Unlike the phage heads that are isometric or prolate, the capsid of RcGTA is oblate with the shortened dimension along the tail axis (Fig. 1a). The structure of the native RcGTA head with imposed fivefold symmetry has been determined to a resolution of 3.6 Å. The organization of the RcGTA capsid is derived from a T = 3 quasi-icosahedral lattice; however, it lacks five hexamers of major capsid proteins in its central part, which results in a shortening of the head (Fig. 1a). The volume for packaging DNA in the oblate head is 35% smaller than that of

the corresponding T = 3 icosahedral head (Fig. 1a, c, d). It has not been previously recognized that the heads of GTAs are oblate; however, published electron micrographs of GTAs produced by other bacteria from the family Rhodobacteraceae indicate that it is a shared feature[20–22]. The oblate head of RcGTA may represent the smallest capsid of tailed phages that can be assembled, because pentamers of capsid proteins forming a T = 1 particle cannot establish the same interactions with the portal complex as those formed by hexamers (Supplementary Fig. 4). The head of native RcGTA contains five concentric layers of density corresponding to the packaged DNA, which are spaced 27.5–29.1 Å from each other (Supplementary Fig. 5). By contrast, the spacing of DNA in the heads of tailed phages and phage particles transducing staphylococcal pathogenicity island SaPI1 is 21–25 Å (refs. [23–25]). Therefore, the DNA density in RcGTA particles is 10–25% lower than that in the heads of *Caudovirales* phages. The reduced capsid size together with the lower DNA density limit the packaging capacity of the RcGTA head to 4000–4500 base pairs, which is less than the 15,000 base pairs that encode the major cluster of structural proteins of RcGTA (Fig. 1b). A particle of RcGTA with a T = 3 quasi-icosahedral head would not be capable of transferring its complete coding sequence; however, it would increase the chance that a combination of a few particles could transfer the whole coding sequence in segments to one recipient. If the RcGTA particles with icosahedral heads acquired the ability to preferentially package their coding sequences, they could become self-propagating and revert to a bacteriophage-like lifestyle. It is possible that the reduced DNA packaging capacity of the oblate heads is a mechanism that prevents the decoupling of the propagation of RcGTA from that of the producer cells.

Horizontal gene transfer mediated by GTAs provides cells with new traits that may increase their fitness[18]. It has been shown that 1000 base pair-long flanking regions of a DNA segment enable a high frequency of homologous recombination in a proteobacterium[26]. The average size of a prokaryotic gene is 1000 base pairs, with 93% of *R. capsulatus* genes being shorter than 2000 base pairs. Therefore, 4000–4500 bases of double-stranded DNA packaged inside the oblate heads of GTAs are well suited for mediating gene exchange in Rhodobacteraceae.

### The major capsid protein of RcGTA.
The 398-residue-long major capsid protein of RcGTA (Rcc01687, g5) has the canonical HK97 fold shared by tailed phages and herpesviruses[27] (Fig. 2a). The quasi-equivalent structure of the T = 3 icosahedral capsid includes conformational differences in the major capsid proteins from one icosahedral asymmetric unit (Supplementary Fig. 6a, c). In the major capsid proteins that form hexamers, the N-terminal arm, core helix from the peripheral domain, and extended loop lie in the same plane (Supplementary Fig. 6c). By contrast, in the major capsid proteins which form pentamers, the structures are bent 18°, 8°, and 10° towards the center of the capsid (Supplementary Fig. 6c).

The formation of the oblate head of RcGTA requires conformational adjustments of quasi-hexamers of major capsid proteins positioned on the twofold axes of the oblate head relative to the quasi-hexamers positioned on the threefold axes of the T = 3 quasi-icosahedral capsid (Fig. 1a, c, d, Supplementary Fig. 6b, d, e). The planar quasi-hexamers in the icosahedral head are formed by capsid proteins in two alternating conformations, depending on whether they bind to pentamers or hexamers of capsid proteins. By contrast, the quasi-hexamers on twofold axes of the oblate capsid are bent to enable seamless closure of the oblate head (Supplementary Fig. 6d, e). The quasi-hexamers on twofold axes of the oblate head contain subunits in three conformations: two subunits binding to pentamers from different

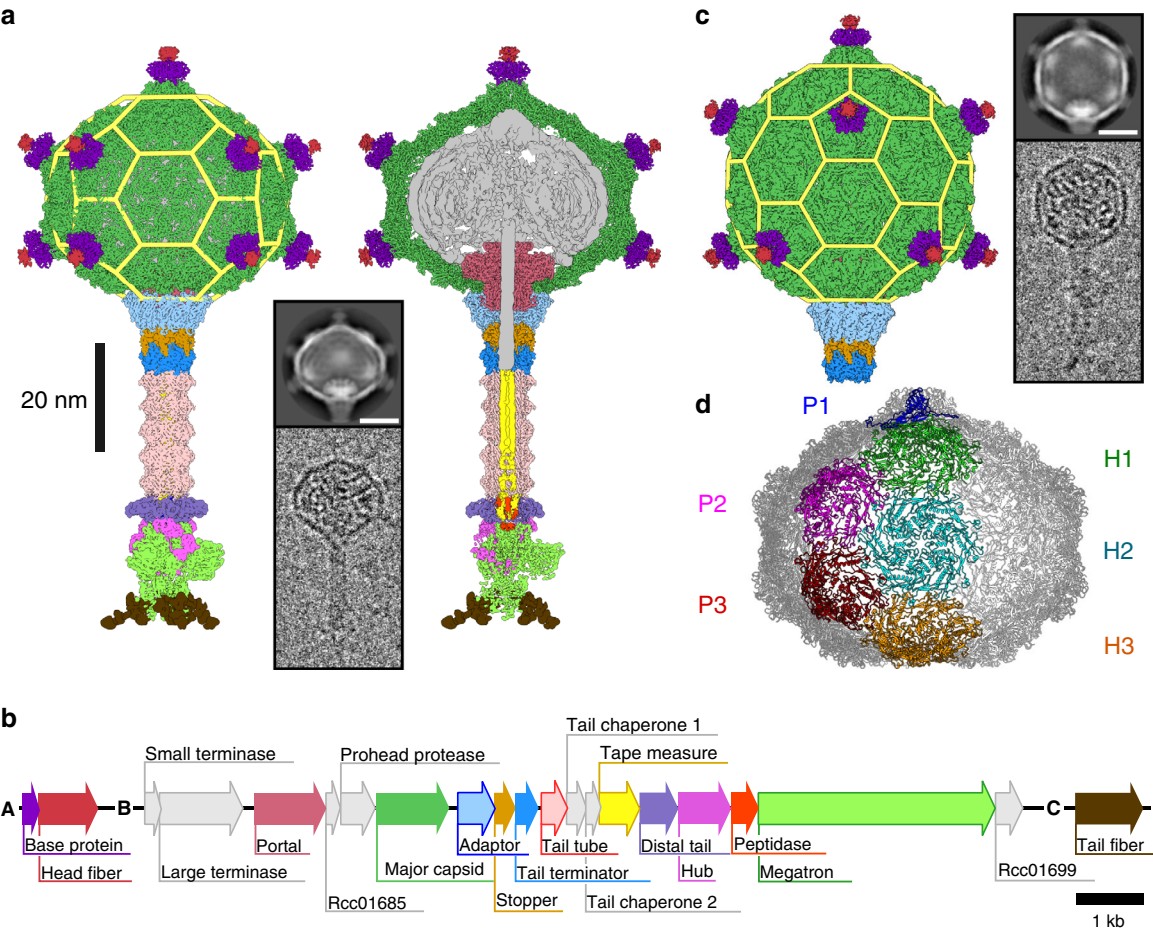

**Fig. 1 Structure of the RcGTA particle and organization of segments of the *R. capsulatus* genome encoding protein components of RcGTA particles.**
**a** Cryo-EM reconstruction of a native particle of RcGTA from *R. capsulatus* strain DE442 calculated from 42,242 particle images. The left part of the panel shows the complete particle, whereas on the right the front half of the particle has been removed to show DNA and internal proteins. Individual proteins in the density map are colored according to the gene map in panel **b**. Yellow mesh highlights the structural organization of capsid proteins within the RcGTA head. The inset shows an example of a two-dimensional class average and an electron micrograph of an RcGTA particle. The scale bar within the inset represents 20 nm. **b** Gene map of three genome segments encoding fourteen structural proteins of RcGTA particles. **c** Cryo-EM reconstruction of an RcGTA particle from *R. capsulatus* strain DE442 with T = 3 quasi-icosahedral head. The reconstruction is based on 1076 particle images. The structure is at the scale of those shown in panel **a**. The inset shows an example of a two-dimensional class average and an electron micrograph of RcGTA particle with an icosahedral head. Scale bar represents 20 nm. **d** Organization of capsomers in the oblate capsid of RcGTA. Capsomers forming one fifth of the capsid are highlighted in different colors and marked with P for pentamer and H for hexamer.

directions, and one binding to a hexamer of capsid proteins (Supplementary Fig. 6b, d). The two pentamer-binding subunits in the oblate head differ from each other in the positioning of their N-terminal arms and extended loops (Supplementary Fig. 6b, c).

**A sub-population of RcGTA particles with icosahedral heads.**
One percent of the population of RcGTA are particles with isometric T = 3 icosahedral heads (Fig. 1c). The structure of the isometric RcGTA capsid with imposed icosahedral symmetry has been determined to a resolution of 4.0 Å (Supplementary Figs. 1–3, Supplementary Table 1). The existence of RcGTA particles with icosahedral heads provides evidence that the assembly of the oblate capsid is not based on intrinsic properties of the RcGTA major capsid protein but may be determined by scaffolding proteins. The locus of genes encoding proteins forming the head of RcGTA includes a hypothetical Rcc01685 with an as-yet unknown function (Fig. 1b, Supplementary Fig. 7a). Rcc01685 has the predicted structure of a 75-residue-long α-helix, similar to that of the scaffolding protein of phage phi29 from the family *Podoviridae*[28]

(Supplementary Fig. 7b–d). However, an *R. capsulatus* knock-out of Rcc01685 (ref. [14]) produces oblate capsids (Supplementary Fig. 7e). Therefore, other proteins must be responsible for determining the head shape of RcGTA.

**Head spikes of RcGTA.** The surface of the RcGTA head is decorated with eleven 70-Å long head spikes attached to pentamers of major capsid proteins (Fig. 1a). Each head spike is composed of a pentamer of base proteins Rcc01079, and a single subunit of head fiber protein Rcc01080 (Fig. 2b–f). The base protein has a jellyroll fold formed by β-strands 1–6. Proteins of similar fold form protrusions at the surfaces of bacterial, archaeal, and eukaryotic viruses (Supplementary Table 4). Base proteins attach to the capsid by the N-termini, each of which binds to axial domains of two adjacent major capsid proteins within a pentamer (Fig. 2b–d). The attachment is reinforced by the coordination of a cation (Fig. 2c). The reconstruction of the head spike only contains resolved high-resolution density for residues 2–10 out of 325 of the head fiber protein (Fig. 2e). The residues **Ile**-Ala-**Leu**-Gly-**Leu**-Gly-**Leu**-Gly-**Leu** form a five-point star bound to the pentamer of base proteins

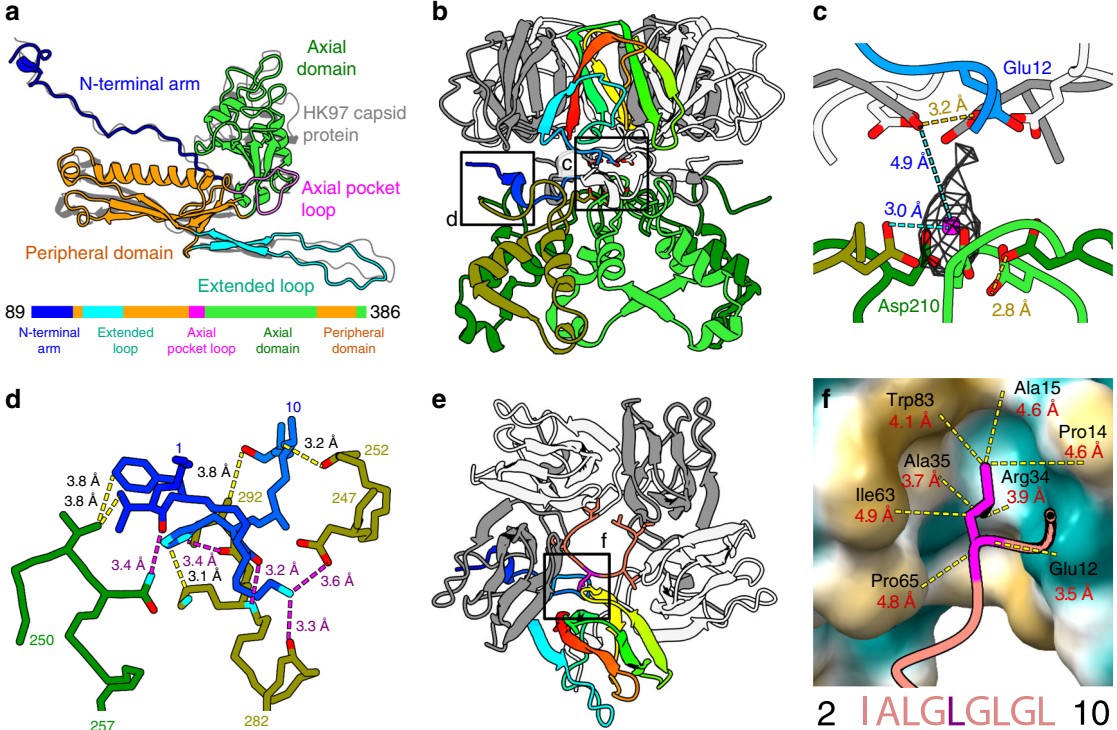

**Fig. 2 Structure of the major capsid protein and head fibers of RcGTA. a** The major capsid protein of RcGTA has the HK97 fold. The structure of the capsid protein of phage HK97 (PDB 1OHG, shown in gray) is superimposed onto that of RcGTA. The domain organization of the protein is shown in the sequence diagram at the bottom of the panel. **b** The interaction of base proteins of the head spike with axial domains of major capsid proteins. One subunit of the base protein is rainbow-colored from the N-terminus in blue to the C-terminus in red; the other four subunits are shown in gray and white. The axial domains of the five major capsid proteins are differentiated by shades of green. Details highlighted with black squares are shown in higher magnification in panels **c** and **d**. **c** Coordination of a putative cation by sidechains of Asp210 of major capsid proteins and Glu12 of base proteins strengthens the attachment of base proteins to the RcGTA head. The sidechains of residues interacting with the putative cation are shown in stick representation. Proteins are colored as in **b**. **d** Detail of the interaction of the N-terminus of the base protein (blue) with two subunits of major capsid proteins differentiated by being colored in olive and green. Selected interatomic distances are indicated by dashed yellow lines, salt bridges are highlighted with magenta lines. **e** Top view of a pentamer of base proteins with attached N-terminus of head fiber protein, shown in salmon. The isoleucine and repetition of the leucine residues enable the binding of the head fiber to the pentamer of head spike base proteins. Detail of the indicated interaction is shown in **f**. **f** Interaction of leucine 6 of the head fiber, shown in magenta, with the hydrophobic pocket of the head fiber base protein, which is shown as a molecular surface. Yellow indicates a hydrophobic surface and turquoise a charged surface. Distances between selected atoms are indicated. The repetitive sequence from the N-terminus of the head fiber protein is shown at the bottom of the panel.

(Fig. 2e). The repeated (iso)leucines enable the peptide to bind to hydrophobic pockets of five base proteins (Fig. 2EF). The rest of the head fiber protein is flexible, as demonstrated in two-dimensional class averages of the head spikes (Fig. 1d, Supplementary Fig. 8). It was shown that RcGTA head fibers enable the attachment of particles to the polysaccharide capsule of *R. capsulatus*[29]. The flexibility of the head fibers may facilitate the polyvalent binding of RcGTA to the capsule.

**Incorporation of the portal complex into the RcGTA head.** The RcGTA head contains a special vertex in which a dodecamer of portal protein subunits (Rcc01684, g3) replaces a pentamer of major capsid proteins (Figs. 1a, 3a). The structure of the portal complex with imposed twelvefold symmetry was determined to a resolution of 3.3 Å (Supplementary Figs. 1–3, Supplementary Table 1). The portal complex is cone-shaped and contains a central channel with a diameter of 28 Å at its narrowest point (Fig. 3a). According to the convention established for the portal proteins of phages[30], the portal of RcGTA can be divided into the clip, stem, wing, and crown domains (Fig. 3a, Supplementary Table 2). The portal of RcGTA is structurally most similar to that of *Thermus* phage G20C from the family *Siphoviridae*[31] (Supplementary Table 2).

Asymmetric reconstruction of the RcGTA particle at a resolution of 4.3 Å enabled characterization of the interface between the capsid and portal complex (Fig. 3b, d, e). Incorporation of the portal complex into the capsid is enabled by changes in the structure of the major capsid proteins relative to their structures in the rest of the capsid. Residues 99–106 from the N-termini of major capsid proteins interacting with the portal are tilted 18° away from the portal complex relative to their orientation in other capsid proteins (Fig. 3e). This conformational change enables the N-terminus to fit inside a groove formed by the wing domain of the portal complex and peripheral domain and extended loop of the capsid protein (Fig. 3E). By contrast, the N-termini of capsid proteins of phage P68 from the family *Podoviridae* bind to the stem domains of portal proteins[32]. The asymmetric reconstruction of the RcGTA particle shows that interactions of the capsid with portal and adaptor complexes induce deviations of the structures from their ideal fivefold and twelvefold symmetries, respectively (Supplementary Fig. 9).

**The adaptor complex mediates reduction of tail symmetry.** A dodecamer of adaptor proteins (Rcc01688, g6) is attached to the surface of the portal complex exposed on the outside of the head and the adjacent part of the RcGTA capsid (Figs. 1a, 3a). The

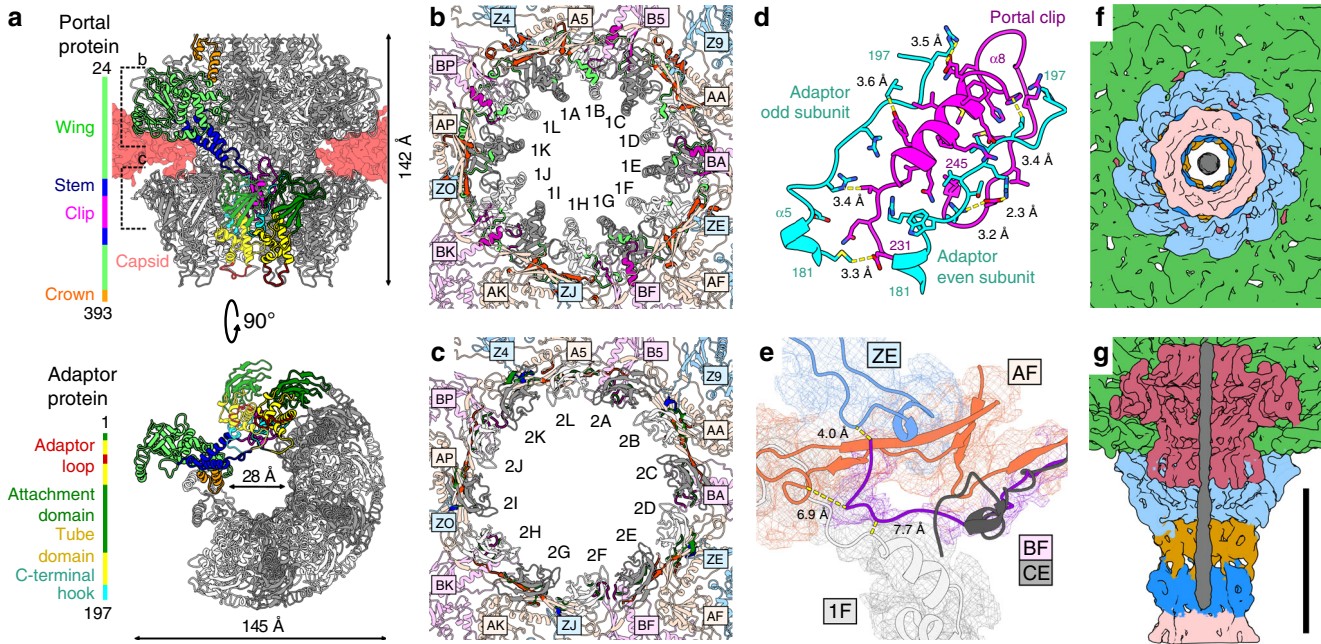

**Fig. 3 Portal and adaptor complexes. a** Side and bottom views of portal and adaptor complexes with one subunit of the portal protein and two subunits of the adaptor protein highlighted in different colors. One of the adaptor proteins is highlighted with a black outline. Five subunits of portal protein and four subunits of adaptor proteins were removed from the bottom view. The domains of the selected subunits are color-coded as indicated in the sequence diagrams on the left. Differences in the structures of adaptor loops (red) of neighboring adaptor proteins enable reduction of the tail symmetry from twelvefold to six-fold. **b, c** Interactions of wing domain of portal proteins (**b**) and attachment domain of adaptor proteins (**c**) with capsid. Alternating subunits of portal (**b**) and adaptor (**c**) complexes are shown in white and gray. Residues of portal and adaptor proteins that interact with the capsid are highlighted in bright colors according to which domain they belong to. The portal and adaptor complexes interact with capsid proteins in three different orientations, which are distinguished by magenta, orange, and blue. Residues of capsid proteins that bind to portal and adaptor proteins are highlighted in bright colors. Subunit are labelled according to PDB 6TBA. **d** Interactions between the clip domain of the portal protein (magenta) and C-terminal hooks of adaptor proteins (cyan). Sidechains of interacting residues are shown in stick representations. **e** Unique structure of the N-terminus of capsid protein subunit BF (magenta) interacting with the portal complex (light gray). Capsid protein CE (gray), which does not interact with the portal complex, was superimposed onto the BF subunit. The N-terminus of the CE subunit clashes with the surrounding structures. **f, g** The end of the double-stranded DNA positioned in the RcGTA neck does not bind to the surrounding proteins of the native RcGTA. **f** View of tail along its axis towards the center of the head. **g** View of neck region with front half removed. DNA is shown in gray, tail tube proteins in pink, tail terminator proteins in blue, stopper proteins in dark orange, adaptor proteins in cyan, portal proteins in purple, and capsid proteins in green. Scale bar 10 nm.

structure of the adaptor complex with imposed twelvefold symmetry was determined to a resolution of 3.3 Å. The maximum outer diameter of the adaptor complex is 143 Å and the minimum inner diameter is 35 Å. The adaptor protein can be divided into four domains: the attachment domain composed of β-strands 1–5, the tube domain formed by α-helices 1–4, the adaptor loop, and the C-terminal hook (Fig. 3a, Supplementary Table 2). Residues 175–197 from the C-terminal hook of the adaptor protein interact with helix α8 from the clip domain of the portal complex (Fig. 3a, d). In addition, the attachment domain of the adaptor protein binds to the α-helix formed by residues 148–152 from the extended loop of the adjacent capsid protein (Fig. 3a, d). By contrast, residues 148–152 of capsid proteins that interact with a pentamer of capsid proteins form a loop. Furthermore, residues from the β3–5 loop of the attachment domain of adaptor proteins interact with five regions of the major capsid proteins (Fig. 3c). The interactions are variable because of the mismatch of the fivefold symmetry of the capsid and twelvefold symmetry of the adaptor complex (Fig. 3c). Phages HK97 and SPP1 from the family *Siphoviridae*, and Mu from the family *Myoviridae* possess adaptor complexes with tube domains and C-terminal hooks similar to that of RcGTA[33–35] (Supplementary Table 4). Phage T7 and its relatives contain adaptor proteins with attachment domains; however, none of them has been shown to bind to a capsid[36].

The dodecamer of the adaptor proteins provides the binding site for a hexamer of stopper proteins, resulting in a symmetry mismatch between the two complexes (Figs. 1a, 4a). Adaptor loops from two neighboring adaptor proteins differ in their conformations and interact with one subunit of stopper protein (Fig. 3a). The first adaptor loop is oriented parallel to the tail axis and fits into a cleft within β-sheets of a stopper protein, whereas the adaptor loop of the second subunit is wedged between two stopper proteins (Supplementary Fig. 10a, b).

**Portal and adaptor of RcGTA do not bind DNA**. It has been shown that subunits of portal or adaptor complexes can bind to an end of the DNA in a bacteriophage head to stabilize its native state[32,37]. The disruption of this interaction was speculated to regulate phage DNA release[32,37]. By contrast, an asymmetric reconstruction of the portal and adaptor complexes of RcGTA, determined to a resolution of 4.3 Å, does not show any unique interactions between the portal or adaptor proteins and the packaged DNA (Fig. 3f, g). Therefore, the DNA is probably held in the RcGTA head by the tape measure proteins and iris-like constriction within the baseplate, which block the tail channel as discussed below.

**Gene multiplication of RcGTA tail proteins**. The central part of the tail of RcGTA is formed by one hexamer of stopper proteins, one hexamer of tail terminator proteins, five hexamers of tail tube proteins, and one hexamer of distal tail proteins positioned in the direction from the head to the baseplate (Figs. 1a, 4a, Supplementary

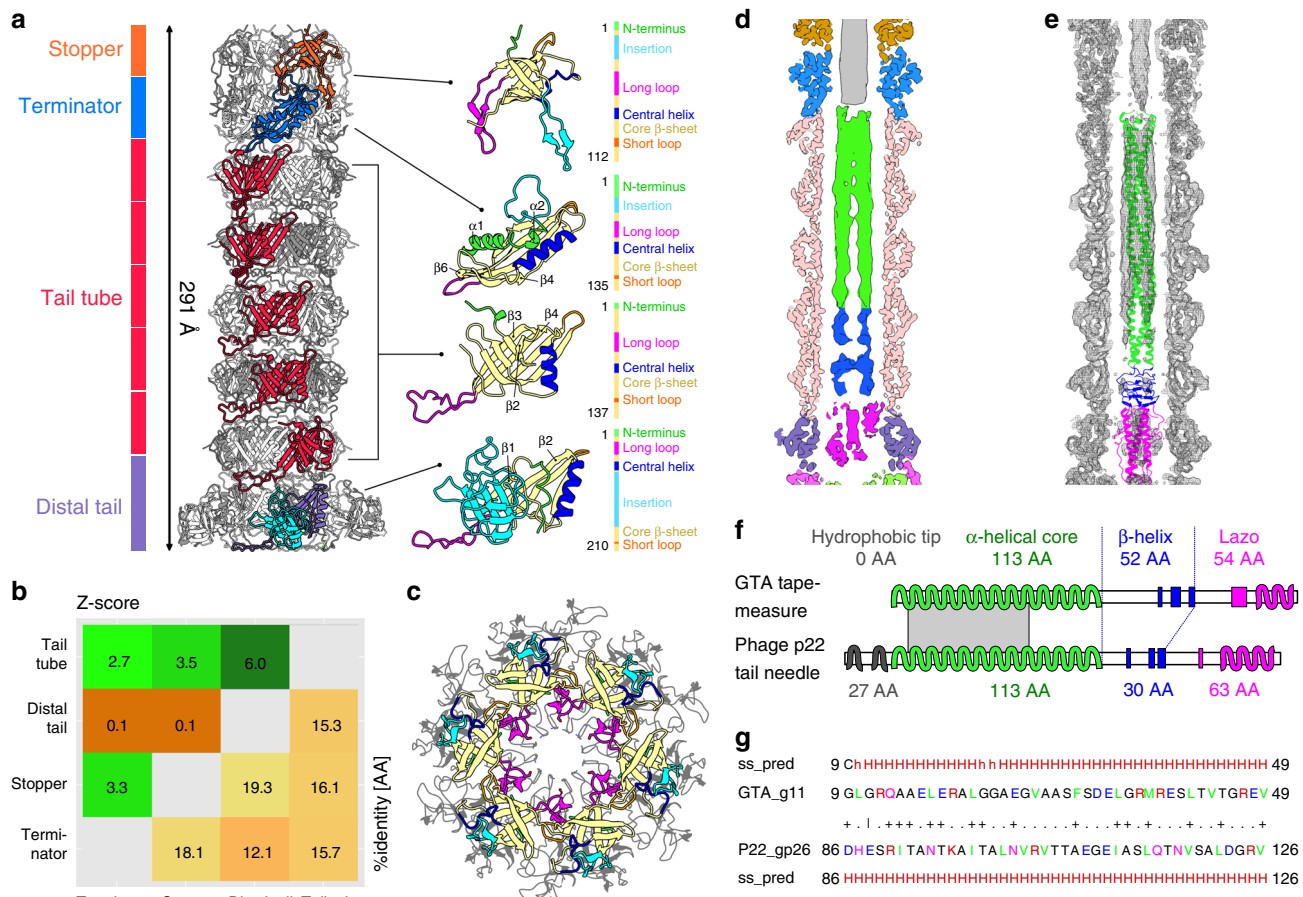

**Fig. 4 Structure of the RcGTA tail. a** The tube of the RcGTA tail is formed by the stopper, tail terminator, tail tube, and distal tail proteins. On the right side, the proteins are shown in cartoon representations with β-strands forming the core of the proteins in yellow, N-termini in green, short loops in orange, long loops in magenta, insertion loops in cyan, and central helices in blue. **b** Structural similarity (upper left) and sequence identity (bottom right) of RcGTA tail proteins. Z-scores were calculated using the DALI server[75]. Values higher than two indicate that the compared proteins are similar. **c** Superposition of hexamers of stopper proteins of RcGTA, colored as in **a**, and phage SPP1 (PDB 5A20_EF), in gray. The sidechains of residues that form the bottlenecks in tails of RcGTA and SPP1 are shown in stick representation. The long loop of the stopper protein of RcGTA (magenta) does not reach as close to the center of the channel as that of SPP1. **d** Central slice through cryo-EM map of RcGTA tail. The parts of the density belonging to RcGTA proteins are color-coded as in panel **a**. The density in the central channel is color-coded according to the domains of tape measure protein shown in panel **f**. **e** Cryo-EM map of RcGTA tail with fitted tail proteins and tail-needle protein of P22 in cartoon representation. The P22 tail-needle model is color-coded according domains shown in panel **f**. **f** Tape measure protein of RcGTA is structurally similar to tail-needle protein (PDB 2POH) of phage P22 from the family *Podoviridae*. Diagrams of secondary structure elements of the two proteins are shown. α-helices are indicated by wiggly lines and β-strands by broad colored lines. The N-terminal region (grey), responsible for the attachment of the needle protein to the tip of P22 tail, is missing from the RcGTA protein. The gray rectangle indicates the position of the sequence displayed in panel **g**. **g** Sequence and secondary structure alignment of 41 residues from coiled-coil regions of the RcGTA tape measure protein and phage P22 tail-needle protein computed using HHpred[44].

Table 5). The structures of the stopper, tail terminator and tail tube proteins have been determined to a resolution of 3.6 Å, and that of the distal tail protein to 4.0 Å (Supplementary Figs. 1–3, Supplementary Table 1). The folds of all RcGTA tail proteins resemble those of tail tube proteins of phages from the family *Siphoviridae*[33] (Fig. 4a, Supplementary Table 2). Although the sequence identity among the RcGTA stopper, terminator, tail tube, and distal tail proteins is less than 19%, the similarities in their overall structures provide evidence of their common origin from one gene (Fig. 4b). The proteins are built from N-terminal α-helix, four to eight core β-strands that form an anti-parallel β-barrel, and several loops of varying[38] (Fig. 4a, Supplementary Table 5).

**Stopper proteins**. A hexamer of RcGTA stopper proteins (Rcc01689, g7) binds to the adaptor complex (Figs. 1a, 4a, Supplementary Fig. 10a, b, Supplementary Table 5). Each 112-residue-long stopper protein interacts with adaptor loops from two

neighboring adaptor proteins (Supplementary Fig. 10a, b). At the distal interface, the long and insertion loops and C-terminus of the stopper protein bind to a terminator protein (Fig. 4a, Supplementary Fig. 10c, d). The stopper protein of RcGTA is named after its homologue gp16 from bacteriophage SPP1 (Fig. 4c). It has been speculated that the long loop of gp16 holds the genome of SPP1 inside the phage head and regulates its release[39]. However, the long loop of the RcGTA stopper protein interacts with the tail terminator protein and does not block the central tail channel (Fig. 4a, c). Furthermore, the end of the DNA packaged in the native RcGTA particle extends through the channel formed by the hexamer of stopper proteins and continues into the disc of tail terminator proteins (Figs. 1a, 3g).

**Terminator proteins**. A hexamer of tail terminator proteins (Rcc01690, g8) binds to the distal interface of the stopper proteins (Figs. 1a, 4a, Supplementary Table 5). The terminator protein is

designated according to its homologue from phage lambda, in which it is the last protein added to the assembling tail before it can be attached to the phage head[40]. Unlike other tail proteins of RcGTA, the tail terminator protein contains α-helices 1 and 2 in its N-terminus. α-helix 1 binds to the insertion loop of the stopper protein and α-helix 2 interacts with α-helix 3 from the same tail terminator protein (Fig. 4a). The insertion and short loops of the tail terminator protein interact with the long and insertion loops of the stopper protein (Supplementary Fig. 10c, d). The distal interface of the tail terminator protein, which is formed by the long loop and strands β4 and β6, provides an attachment site for tail tube proteins (Supplementary Fig. 10e, f).

**Tail tube proteins do not change upon DNA release.** The RcGTA tail contains five discs of tail tube proteins (major tail proteins) (Rcc01691, g9), which are organized as a six-entry helix with a twist of 24.4° and pitch of 38.3 Å (Figs. 1a, 4a). The N-terminus, short loop, and loop β2-β3 of the tail tube protein enable its binding to tail terminator proteins (Fig. 4a, Supplementary Fig. 10e, f, Supplementary Table 5). The long loop of the tail tube protein mediates interactions between the tail tube proteins from successive discs and between the tail tube protein and distal tail protein (Supplementary Fig. 10g, h). The structure of tail tube did not reveal any changes after DNA ejection (Supplementary Fig. 11), suggesting that the tail is not involved in signaling to trigger DNA ejection after attachment to a cell.

**Distal tail proteins.** The distal tail protein (Rcc01695, g12) was named after its homologue from phage T5 (ref. [41]) (Fig. 4a). It is similar to the tail tube protein but contains an extra insertion domain (Fig. 4a, Supplementary Table 5). The attachment of distal tail proteins to tail tube proteins is enabled by the long loop of the tail tube protein, which interacts with the β2-core α-helix loop and short loop of the distal tail protein (Fig. 4a, Supplementary Fig. 10i, j). The insertion domain of the distal tail protein was resolved to a resolution of 5 Å and can adopt several conformations, as seen in two-dimensional class averages of RcGTA tails (Supplementary Fig. 12). The insertion domain of this RcGTA protein is homologous to that from the distal tail protein of phage T5, which has an oligosaccharide-binding fold[41]. Therefore, it is possible that the insertion domain enables the binding of RcGTA particles to a sugar receptor at the cell surface of *R. capsulatus*, and flexibility of the domain may facilitate binding.

**Tape measure protein.** The part of the tail channel of RcGTA formed by the tail tube and distal tail proteins is filled by a trimer of tape measure proteins (Rcc01694, g11) (Figs. 1a, 4d). The reconstruction of the tape measure protein was determined to a resolution of 5.0 Å. Building an atomic model was not possible due to the limited resolution of the map. However, the cryo-EM density in combination with sequence-based secondary structure prediction provide evidence that the N-terminal part of the RcGTA tape measure protein forms a 113-residue-long α-helix followed by 83 residues of β-strands and loops, and 20 residues of the C-terminal α-helix (Fig. 4e–g, Supplementary Fig. 13). The N-terminal α-helix of the RcGTA tape measure protein contains 12- and 13-residue-long repeats, starting with amino acids containing large sidechains that are characteristic for tape measure proteins of phages from the families *Siphoviridae* and *Myoviridae*[42] (Supplementary Fig. 13c, d). Nevertheless, the content of secondary structure elements and the ability to form rod-shaped trimers of the RcGTA tape measure protein resemble those of tail-needle protein gp26 of phage P22 from the family *Podoviridae*[43] (Fig. 4e–g). Sixty-six residues from the predicted α-helix of g11 of RcGTA can be aligned to residues of gp26 of phage P22 with an

e-value of 0.007 and similarity score of 21%, as determined using the program HHpred[44]. This indicates that tape measure proteins of long-tailed phages and tail-needle proteins of short-tailed phages may have originated from a common precursor.

**RcGTA baseplate.** The structure of the RcGTA baseplate with imposed threefold symmetry has been determined to a resolution of 4.0 Å. The threefold symmetry makes the RcGTA baseplate distinct from those of phages studied to date, which are organized with six-fold or quasi-six-fold symmetries[45–47] (Figs. 1a, 5a). The core of the RcGTA baseplate, formed by the hub (Rcc01696, g13) and multi-domain protein designated megatron (Rcc01698, g15) proteins, is decorated with tail fibers. The hub protein can be divided into attachment (residues 1–142), ion-binding (143–167, 250–263), oligosaccharide-binding (168–249), and clip domains (264–296) (Fig. 5a). The oligosaccharide-binding domain of the RcGTA hub protein was so named because of its similarity to the insertion domain of tail spike protein gp49 of phage LKA1 from the family *Podoviridae*, which was shown to bind sugars[48] (Supplementary Table 2). The iron-binding domain contains four conserved cysteines, which coordinate the iron–sulfur cluster[49] (Fig. 5b).

The structure of the megatron protein is composed of iris/penetration (residues 1–46), adhesin-like (47–229), peripheral (230–744), central (745–984), and fiber-binding (985–1304) domains (Fig. 5a). The iris/penetration domain of the megatron protein contains α-helix 1 (residues 7–16), an extended loop (17–21), disordered region (22–37), and α-helix 2 (38–46). Helices α1 from three megatron proteins form an iris-like constriction that blocks the central channel of the RcGTA tail (Fig. 5c, d). The sequence of the iris/penetration domain indicates that it could form a pore-lining helix[50], which may enable the translocation of DNA from RcGTA particles across the outer membrane of *R. capsulatus* (Supplementary Fig. 14).

Attachment of the RcGTA baseplate to the tail is enabled by the binding of the attachment domain of the hub protein and central domain of the megatron protein to the N-terminal Ala2, long loops, and C-terminal Arg209 of the distal tail proteins (Supplementary Fig. 15). The mismatch between the six-fold symmetry of the tail and threefold symmetry of the baseplate is resolved by the different conformations of residues from the long loops of odd and even subunits of distal tail proteins, which enable them to interact with hub and megatron proteins, respectively (Supplementary Fig. 15b–d).

The trimers of hub and megatron proteins form a compact complex with a buried surface area of the interaction interface of 3550 Å². Comparison of the structures of RcGTA baseplate proteins with those of phages provides evidence of domain swapping. The hub protein (gp27) of phage T4 from the family *Myoviridae* and the VgrG1 protein of the type VI secretion system of *Pseudomonas aeruginosa* contain domains homologous to the attachment domain of the hub protein and central domain of the megatron protein of RcGTA[47,51] (Supplementary Fig. 16). Furthermore, the oligosaccharide-binding domain of the hub protein of RcGTA resembles that of the bacteriophage T4 hub protein (Fig. 5e, Supplementary Fig. 16). Although the proteins of RcGTA, T4, and the secretion system share less than 19% sequence identity, the domains can be superimposed with an RMSD of the corresponding atoms of less than 3.8 Å.

**Tail fibers.** Tail fibers of RcGTA are thought to bind to receptors at the surface of *Rhodobacter* cells and are essential for the gene transfer activity of the particles[52]. The cryo-EM map of the RcGTA tail fiber (Rcc00171) was determined to resolutions of 6.8 Å and 13.9 Å for the parts that are proximal and distal to the baseplate, respectively (Fig. 5a, Supplementary Fig. 17a). The

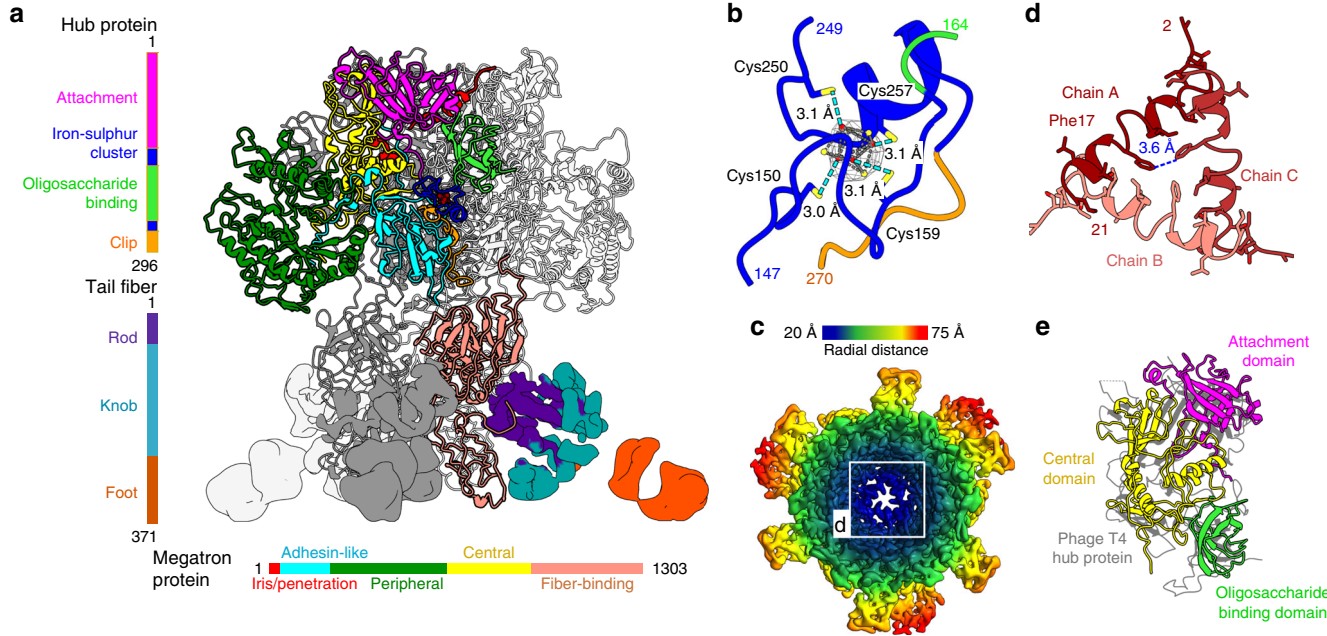

**Fig. 5 Structure of the RcGTA baseplate. a** Side-view of the RcGTA baseplate. Domains of one hub and one megatron protein are colored according to the sequence diagrams shown on the left and at the bottom of the panel. The iron–sulfur cluster in the hub protein is shown as dark red spheres. Electron densities of tail fibers are shown. The density of one of the fibers is colored according to domains as indicated in the sequence diagram on the right of the panel. **b** Detail of iron–sulfur cluster coordinated by four cysteines of hub protein. The electron density map of the cluster is stronger than that corresponding to the surrounding proteins. Distances between sulfur atoms of cysteine sidechains (yellow) and iron ions (red) are indicated. **c** Cryo-EM map of the RcGTA baseplate viewed along its axis towards the head is rainbow-colored based on the distance from the threefold axis of the structure. The inset shows detail of the constriction of the central channel formed by iris/penetration domains of megatron proteins. **d** The iris/penetration domains from three subunits of megatron proteins are differentiated by shades of red. Interatomic distances between sidechains of Phe17 are indicated. **e** Domain swapping among baseplate proteins of RcGTA and phage T4. Attachment and oligosaccharide-binding domains of the RcGTA hub protein and the central domain of the megatron protein of RcGTA can be superimposed onto the central hub protein of bacteriophage T4 shown in gray (PDB 1K28).

structures of homologs of the RcGTA fiber from R1-pyocin (PDB 6CXB) and phage AP22 (PDB 4MTM) could be fitted into the reconstructed density[53,54] (Supplementary Fig. 17b). Following the nomenclature established for R-type pyocins[55], the 371-residue-long tail fiber protein of RcGTA can be divided into an N-terminal α-helical rod domain (residues 1–45), knob domain (46–259), and C-terminal β-propeller foot domain (260–371) (Supplementary Fig. 17b, c). The similarity of the predicted distribution of secondary structure elements of the foot domain of the RcGTA tail fiber to that of the lectin domain of the tail fiber of R1-pyocin provides additional evidence that the receptor recognized by RcGTA tail fibers is a sugar, as previously speculated by Hynes et al.[52]. The fiber-binding domain of the megatron protein, which provides an attachment site for the fiber protein, is held in position by a 20-residue-long linker and interaction with the peripheral domain of the megatron (Supplementary Fig. 18). Three-dimensional classification identified sub-groups of RcGTA baseplates with flexible tail fibers that lack the connection between the fiber binding and peripheral domains of the megatron protein (Supplementary Fig. 18). The movements of tail fibers relative to the baseplate may increase the probability of their binding to a receptor.

**Tail peptidoglycan peptidase.** The central channel of the RcGTA tail contains a fragmented density located between the iris-like constriction formed by the penetration domains of megatron proteins and the C-terminal domains of tape measure proteins (Fig. 1a, Supplementary Fig. 18a). The classification of electron micrographs of RcGTA particles showed that this density is also present in aberrant particles lacking the tape measure protein (Supplementary Fig. 18a, b). The volume of the density

corresponds to the molecular mass of a single monomer of the 150-residue-long peptidase (Rcc01697, g14), which was shown to be capable of degrading peptidoglycan from *R. capsulatus* cells[11]. Attempts to calculate an asymmetric reconstruction of the peptidase were unsuccessful, probably due to its small mass. The deletion of the peptidase gene prevented the formation of native particles of RcGTA[11], and we observed empty heads without tails (Supplementary Fig. 19). Therefore, it cannot be determined with certainty that the unassigned density in the RcGTA tail belongs to RcGTA peptidase g14.

**Reorganization of baseplate regulates DNA release.** The tail channel of the RcGTA native particle is constricted by an iris made from α-helices 1 of megatron proteins (Fig. 5c, d). The iris has to open to enable genome ejection. The native baseplate of RcGTA does not contain the space necessary to shift the α-helices 1 away from the pore (Fig. 5a, c, d). Therefore, the release of DNA from the RcGTA particle requires re-arrangement of the baseplate, which is consistent with the observation that one third of empty RcGTA particles lacked baseplates (Supplementary Fig. 18b). The central channel of the RcGTA tail above the iris contains a trimer of tape measure proteins and perhaps also one molecule of peptidoglycan peptidase (Fig. 1a, Supplementary Fig. 13e, f). These proteins have to be released from the virion before the DNA can exit. Some of the empty particles with the baseplate still present contained density corresponding to the inner tail proteins, which would not allow the DNA ejection through the tail (Supplementary Fig. 18b). We speculate that the empty particles with attached baseplate are defective and never contained a full complement of DNA.

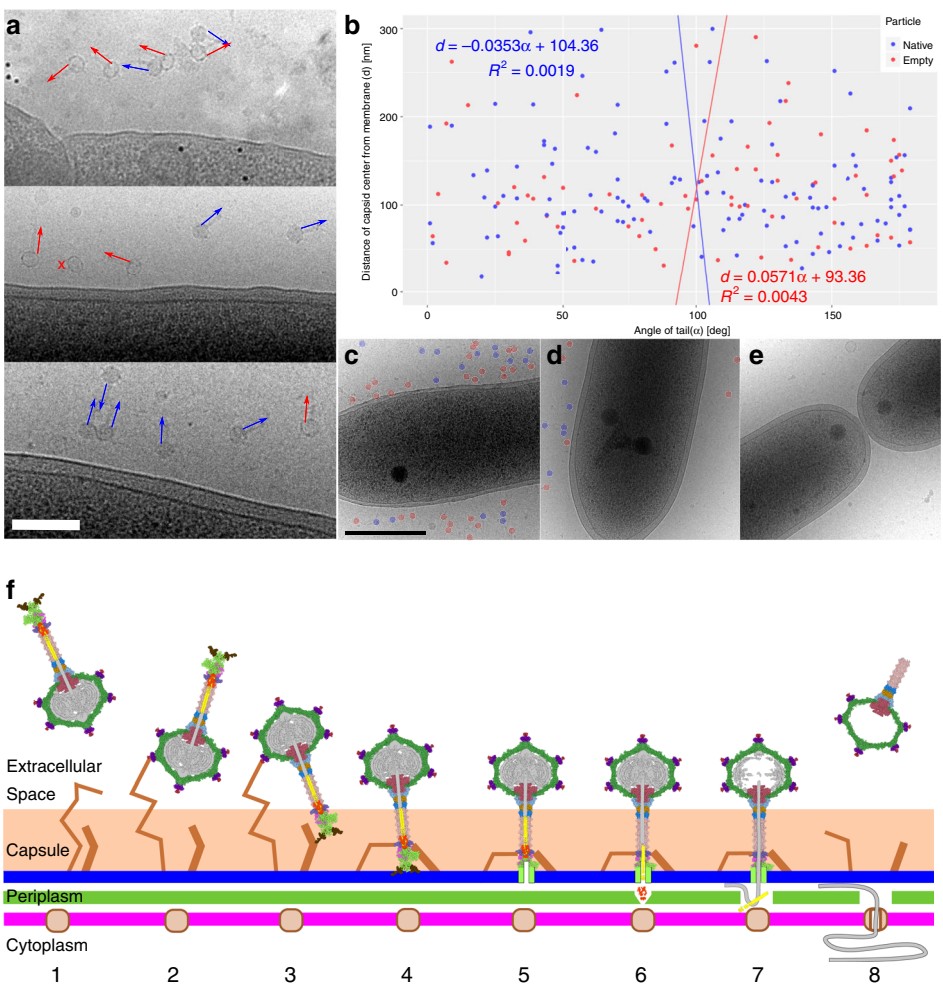

**Fig. 6 Mechanism of DNA delivery by RcGTA. a, b** RcGTA particles attach to cells in random orientations. **a** Cryo-electron micrographs of particles of RcGTA attached to cells of *R. capsulatus*. The panel includes images from three biological replicates. Orientations of RcGTA tails are highlighted with arrows. Blue indicates genome-containing particles and red empty ones. A cross next to a particle indicates that its tail is not visible in the projection image. Black dots are fiducial markers. Scale bar represents 200 nm. **b** Distribution of orientations of tails of RcGTA particles (*x*-axis) versus distance of capsid center from outer cell membrane (*y*-axis). The tail orientation has values from 0°, when the tail points at the membrane, to 180°, when the tail points away from membrane. The particles are oriented randomly. **c–e** Cells of *R. capsulatus* are heterogeneous in their capacity to bind RcGTA. Electron micrographs of three *R. capsulatus* cells from the same experiment using multiplicity of Rif-transferring RcGTA of 0.0002. Some cells were covered with numerous RcGTA particles (**c**), some had tens of them attached (**d**), whereas the remaining ones only attracted a few (**e**). Native and empty RcGTA particles are highlighted with blue and red circles, respectively. **f** Model of RcGTA-mediated DNA delivery. (1) Free particle. (2) RcGTA attaches to the cell capsule by the head fibers. (3) Particle reorients by the binding of tail fibers to outer membrane receptors. (4) Particle attaches to the membrane by putative receptor-binding domains of the baseplate. (5) Penetration of the outer membrane by iris/penetration domain of megatron protein. (6) Ejection of cell-wall peptidase into periplasm enables degradation of cell wall. (7) Ejection of tape measure protein with DNA to periplasmic space. (8) Uptake of DNA by cell competence system.

**Attachment and DNA delivery mechanism of RcGTA.** Based on the structures of native and empty particles of RcGTA and cryo-EM images of RcGTA attached to *R. capsulatus* cells (Fig. 6a, b), the DNA delivery mechanism can be proposed (Fig. 6f). The head of RcGTA is decorated with eleven head spikes, which contain oligosaccharide-binding sites and mediate the initial reversible attachment of RcGTA to the capsule of the *R. capsulatus* cell (Fig. 6f). Cells of *R. capsulatus* are heterogeneous in their capacity to bind RcGTA, since numerous RcGTA particles bind to some cells, whereas others do not attract any (Fig. 6c–e). The tail of RcGTA contains twenty-one oligosaccharide-binding sites, six of which are within the distal tail proteins, three in hub proteins, three in megatron proteins, and nine in tail fiber proteins (Supplementary Fig. 17c). Dynamic binding of the various receptor-binding sites may enable the passive penetration of RcGTA particles through the 40–130-nm thick capsule so that the

particles reach the bacterial outer membrane (Fig. 6a). For productive DNA ejection, RcGTA particles must orient with their baseplate towards the outer membrane, presumably facilitated by binding of tail fibers to an outer membrane receptor (Fig. 6f). The receptor binding of hub and megatron proteins may lead to destabilization of the baseplate, resulting in an opening of the iris within the baseplate and exposure of the N-termini of the megatron proteins. The N-termini of megatron proteins probably insert into the outer membrane of the *R. capsulatus* cell and form a transmembrane pore[50]. Conformational changes of the baseplate trigger the release of inner tail proteins, probably also including the tail peptidase, which degrades peptidoglycan in the periplasm of *R. capsulatus*. The tape measure proteins and DNA packaged in RcGTA head are subsequently ejected into the periplasm. It was shown that the DNA can remain in the bacterial periplasm for several hours until it is imported into the cytoplasm

by homologues of transformation-competence proteins of the recipient *R. capsulatus* cells[15].

## Methods

**Production of RcGTA particles**. The overproducer strain of *R. capsulatus* DE442 was used for the preparation of RcGTA particles. Mutants of *R. capsulatus* strain SB1003 were used for the production of RcGTA particles with knocked-out orf3.5 (ref. [14]) and orf14 (ref. [11]). An overproducer strain of *R. capsulatus* SB1003Δg14 (contains a knocked-out orf14) was created by the RcGTA-mediated transfer of kanamycin resistance-disrupted *rcc00280* from strain SBT2-C22 (refs. [11,12]). Cells from glycerol stock were inoculated into 20 ml of RCV medium[56,57] and incubated for 24 h at 35 °C with 200 rpm shaking. Subsequently, YPS medium was inoculated with 1% by volume of the culture in RCV medium and incubated at 35 °C in sealed 15 ml screw-cap glass tubes kept stationary for 72 h at 38 cm from four 30 W light-emitting tubes (Osram). Lysis connected to the production of RcGTA was monitored by measuring the absorbance of the supernatant at wavelengths from 750–900 nm, looking for the presence of peaks corresponding to the released intracellular LH2 pigment[11].

**Purification of GTA particles**. Cells and cell debris were removed by centrifugation at 8000 × *g* for 30 min, and the supernatant was consecutively filtered through 0.8 and 0.4 um filters. The filtered supernatant was mixed with buffer A (50 mM K-phosphate, pH 7.0; 10 mM NaCl; 5 mM MgSO₄) in the ratio 1:2 (v:v) and applied to a pre-equilibrated CIM-multus QA 8 ml column (BIAseparations)[58]. Particles of RcGTA bound to the column were washed by applying 12 column volumes of buffer A followed by a linear gradient of elution buffer B (50 mM K-phosphate, pH 7.0; 1.8 M NaCl; 5 mM MgSO₄) until the conductivity of the mixture reached 32 mS (~14% (v/v)). The elution of RcGTA particles was induced by an increasing concentration of buffer B until the conductivity reached 40 mS (~16% (v/v)). The presence of native GTA particles was confirmed by agarose gel electrophoresis. Samples (20 µl) of selected fractions were loaded on EtBr-stained agarose gel, separated under constant voltage (5 V × cm⁻¹) and visualized using a UV transilluminator. With the SB1003 strain, the agarose gel electrophoresis step was omitted. Fractions containing native particles of RcGTA were pooled. The mixture was subjected to buffer exchange using Amicon 100 kDa cutoff concentrators to G-buffer (10 mM Tris, pH = 7.8; 1 mM NaCl; 1 mM CaCl₂; 1 mM MgCl₂; with the addition of bovine serum albumin V (Sigma) to a concentration of 10 µg × ml⁻¹)[56]. The concentrated particles were applied to a preformed 15–40% (w/v) sucrose gradient prepared using Gradient station (Biocomp). Gradients were centrifuged at 100,000 × *g* overnight at 8 °C in an SW41Ti rotor (Beckman Coulter) and fractionated using Gradient station (Biocomp). The presence of RcGTA particles in fractions was confirmed by agarose gel electrophoresis. To remove sucrose from the sample, buffer exchange to the G-buffer was performed using Amicon 100 kDa cutoff centrifugal units.

**Cryo-EM sample preparation and data acquisition**. Purified GTA particles were concentrated to 2 mg × ml⁻¹. The concentration of RcGTA particles was determined by measuring the absorbance at 280 nm, assuming an extinction coefficient of 7.7 ml/cm × mg. A sample of concentrated particles (3.9 ul) was applied to glow-discharged holey carbon-coated copper grids (R2/1 300 mesh, Quantifoil), blotted and plunge-frozen in liquid ethane using an FEI Vitrobot Mark IV. Grids were transferred to an FEI Titan Krios TEM operating at 300 kV, where the samples were kept under cryogenic-conditions. Automatic data acquisition was performed using EPU software. The focus range was set to −1 to −3 µm. Imaging was done under low-dose conditions with a total dose of 42.75 e⁻ × Å⁻². Nominal magnification was ×75,000, resulting in a calibrated pixel size of 1.063 Å. Micrographs were collected using a Falcon 3EC direct electron detector operated in linear mode. Micrographs were acquired as 39 fraction movies. Fractions were aligned and dose-weighted using MotionCorr2 (ref. [59]), and the resulting micrographs were used for further processing. Defocus of the micrographs was estimated using gCTF[60].

**Reconstruction of native and empty RcGTA particles**. Subsets of 1000 full and 1000 empty particles of RcGTA were manually picked using EMAN2 e2boxer.py[61]. Particles were extracted with a box size of 512 px, and 2D classification was computed using the program RELION2.1 (ref. [62]). The best-looking class averages were used as templates for autopicking from the whole data set using the program RELION2.1. The parameters of autopicking were optimized on a subset of 20 micrographs. After autopicking and extraction of the particles, the particles were binned twice using Xmipp[63]. Multiple rounds of 2D classification were performed to separate full, empty and icosahedral particles using the program RELION2.1. An initial model with imposed C5 symmetry was generated de novo using the stochastic gradient descent method[64] implemented in RELION2.1. Three-dimensional refinement was performed using 3dautorefine in RELION 3 (ref. [65]) followed by three-dimensional classification where the orientational search was omitted (skip_align option), and the particles were classified into four classes. Particles forming the best class were selected for the final reconstruction of binned particles. The reconstruction of the native particle was computed using a mask excluding the density inside the head, whereas the reconstruction of empty particles was performed without masking. The mask was prepared using volume segmentation in

UCSF Chimera[66] and the relion_mask_create routine from RELION2.1. After the quality and resolution of the binned images stopped improving, the data were unbinned and the final refinement was performed with local angular searches around the known orientation from the binned refinement. To further improve the resolution, CTF refinement was performed using RELION3. The final refinement was done using the skip_align option in 3dautorefine in RELION3. The final resolution was estimated using the FSC₀.₁₄₃ criterion.

**Extraction of sub-particle images from micrographs**. To enable the extraction of parts of the GTA particles (portal/neck region, tail, and baseplate), the particles were re-extracted with a box size of 1440 px. The re-extracted particles were binned four times and C5 reconstruction of the particle was performed. This reconstruction oriented the capsid with its tail on the Z-axis (rotational symmetry axis). Vectors describing the distance of the complexes selected for sub-particle reconstructions from the center of the particle were used for sub-particle extraction using the "localized reconstruction script"[67]. These images were suitable for de novo reconstructions. The box size used for extracting the neck and baseplate was 300 px, and 256 px for the tail. Head spikes were extracted from capsids in 512 px boxes using the option align_subparticles. Two-dimensional classification of the spikes was performed. Further processing of head spikes did not lead to a higher resolution of 3D reconstruction.

**Reconstruction of portal/neck region**. The reconstruction of the portal and neck region was initiated by a refinement with a full search of rotation angle and local searches of tilt and psi angles on twice-binned images. Subsequently, three-dimensional classification and final refinement were performed using unbinned data. Because of the symmetry mismatch between the capsid and portal/neck region, a mask excluding the capsid was applied. For the reconstruction of the portal and adaptor complexes with C12 symmetry, the neck with C6 symmetry was also masked out. Three-dimensional classification with the skip_align option and C6 symmetry, based on the orientations from the C12 reconstruction, was performed to obtain two separate reconstructions of the tail rotated 30° relative to each other. Orientations of particles from one of the groups were adjusted by the addition of 30° to their rotation angle. Where the adjusted value of the rotation angle exceeded 30°, we subtracted 60° from its value to restrict the rotation angle values to the range of −30° to +30° of C6 symmetry. The reconstruction continued with refinement with local searches on the twice-binned data, three-dimensional classification and the final refinement of unbinned data.

**Reconstruction of baseplate region**. Particles with damaged, overlapping, and missing baseplates were excluded by two-dimensional classification. Class averages from the two-dimensional classification indicated that the baseplate has C3 symmetry. Refinement with applied C3 symmetry, a full search of rotation angle and local searches of tilt and psi angle on twice-binned data were performed, followed by three-dimensional classification and final refinement using unbinned data. Asymmetric reconstruction (C1) of the baseplate was performed by local searches of all angles, estimated from the final asymmetric reconstruction of the portal-capsid interaction (see below). To obtain a high-resolution reconstruction of distal tail protein C6, a symmetrized reconstruction of the end of the tail was computed by masking out the baseplate with C3 symmetry and performing a refinement with the skip_align option and known particle orientations from the final C3-symmetrized baseplate reconstruction.

**Helical reconstruction of GTA tail**. The C3-symmetrized three-dimensional reconstruction of the RcGTA baseplate included two discs of tail tube proteins. The resolution of the map was sufficient to determine the helical parameters of the tail tube. For the native virions, a C3-symmetrized refinement using 3dautorefine and a smooth cylinder as the initial model resulted in a low-resolution structure of the tail tube that served as an initial model for further refinement applying C6 and helical symmetries. Reconstruction of the tail of empty particles was initiated based on the orientations derived from the reconstruction of the neck region. Initial refinement employed C6 symmetry and only local angular searches. Subsequent refinements employed helical symmetry. All the refinements were performed using the 3Dautorefine routine from RELION3. The helical parameters for the tail tubes of both native virions and empty particles were 24.4° twist and 38.3 Å pitch.

**Asymmetric reconstruction of portal-capsid interface**. The orientations of capsids used for C5 reconstruction were symmetry-expanded, and values of rotation angles that agreed with the symmetry-expanded rotation values of the C6 refined neck region were selected using a python script (written by Jiří Nováček, CEITEC Cryo-Electron Microscopy and Tomography Core Facility). Then, asymmetric three-dimensional refinement was performed using local angular searches. Three-dimensional classification with the skip_align option was performed. In addition, an asymmetric map of the neck was computed by a method similar to that used for phage p68 (ref.[32]): particle orientations from the final refinement were symmetry-expanded and non-symmetrized (C1) three-dimensional classification was performed, omitting the orientation search, and separating the data set into 12 classes.

**Model building**. Molecular structures were built manually using the software Coot[68]. After building the initial models, the map was zoned in UCSF Chimera by applying a 4 Å mask on the main chain and the model was iteratively refined in real space using the program Phenix[69]. Models were subsequently refined using NCS constraints and interacting partners in the virion, to prevent inter-molecular atom clashes. During the iterative refinement process, the molecular geometry was monitored by MolProbity[70] and geometrical outliers were fixed manually using the program Coot. Unique chains of the final model were selected and symmetry-expanded in Chimera according to the symmetry of the reconstructed map and deposited in PDB. The models of head spike fiber, peptidase and peripheral and fiber-binding domain of megatron protein were computed using RaptorX contact-dependent modelling[71].

**Identification of structural proteins**. Purified GTA particles (50 µl, 2 mg × ml$^{-1}$) were mixed 4:1 with home-made SDS-PAGE loading buffer (0.175 M Tris, pH = 6.8; 15% (w/v) glycerol, 5% (w/v) SDS, 4.7% (w/v) dithiothreitol, 0.04% (w/v) bromophenol blue), heated for 10 min at 95 °C and the proteins were separated on 10–18% Tris-Glycine gradient gel. Proteins were stained with Blue silver[72]. Bands were cut and analyzed by mass spectroscopy at the CEITEC Proteomics Core Facility.

**Bioinformatic analysis and figure preparation**. The functions of individual proteins encoded by the GTA cluster were estimated based on similarities to proteins with known functions, identified by a secondary structure alignment performed in HHpred[73] and primary sequence alignment performed in BLAST[74]. Structures of RcGTA proteins were aligned to published structures using the DALI server[75]. Figures were prepared in UCSF Chimera[66] and UCSF ChimeraX[76]. Multiple sequence alignment of short peptides was performed in Promals3D[77] and visualized in MView[78]. Sequence identity between tail proteins was determined using the program LALIGN with the BLOSUM50 matrix and global alignment method ignoring end-gap penalty[79]. The tail orientation chart, heatmaps, trans-membrane and FSC graphs were prepared using the package R [url https://www.r-project.org/].

**Quantification of attachment of RcGTA to *R. capsulatus* cells**. *R. capsulatus* strain B10 was grown overnight in RCV medium to OD$_{600nm}$ of 0.5–1.0 (ref. [57]). The cells were harvested by centrifugation at $3000 \times g$ for 10 min, the supernatant was discarded and the pellet was resuspended in the same volume of G-buffer. This step was repeated four times. The pellet from the last step was resuspended in G-buffer to obtain a final OD$_{600nm}$ of 32. DNase was added to the sample to a final concentration of 1 µg × ml$^{-1}$, and the sample was incubated for 30 min at room temperature. Subsequently, 4 µl of cell suspension was mixed with 1 µl of purified RcGTA particles in G-buffer at A$_{280nm}$ = 1.5 (corresponding to an MOI of Rif-transferring GTA of 0.0002, measured using a gene transferring assay[80]) and incubated for additional 5–10 min. A sample volume of 3.9 µl was then applied onto glow-discharged holey carbon-coated copper grids (R2/1 200 mesh, Quanti-foil), blotted and plunge-frozen and analyzed in an electron microscope operated at 200 kV. For the statistical evaluation of tail orientations, seven micrographs were analyzed from each grid of a biological triplicate (21 micrographs in total), obtaining data from 74 empty and 125 native particles.

**Reporting summary**. Further information on research design is available in the Nature Research Reporting Summary linked to this article.

## Data availability

Cryo-EM electron density maps have been deposited in the Electron Microscopy Data Bank, https://www.ebi.ac.uk/pdbe/emdb/ (accession numbers are listed in Supplementary Table 1), and the fitted coordinates have been deposited in the Protein Data Bank, www.pdb.org (PDB ID codes are listed in Supplementary Table 1). The authors declare that all other data supporting the findings of this study are available within the article and its Supplementary Information files or are available from the authors upon request.

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

## Acknowledgements

We acknowledge Cryo-electron Microscopy and Tomography, Biomolecular Interactions and Crystallization, and Proteomics core facility of CEITEC supported by MEYS CR (LM2018127). We gratefully acknowledge the support of NVIDIA Corporation for the donation of the Titan Xp GPU used for this research. We wish to thank Dr. Andrew S. Lang for providing *R. capsulatus* SB1003 knock-out strain of rcc01685. This research was carried out under the project CEITEC 2020 (LQ1601), with financial support from the MEYS of the Czech Republic under National Sustainability Program II. This work was supported by IT4I project (CZ.1.05/1.1.00/02.0070), funded by the European Regional Development Fund and the national budget of the Czech Republic via the RDI-OP, as well as the MEYS via the Grant (LM2011033). The research leading to these results has received funding from the Czech Science Foundation grant 18-13064 S to R.P., Masaryk University MUNI/E/0530/2019 to P.B., and Czech Science Foundation grant 18-17810 S and from EMBO installation grant 3041 to PP. The research of J.T.B. was supported by a grant (RGPIN 2018-03898) from the Canadian Natural Sciences and Engineering Research Council (NSERC).

## Author contributions

P.B., R.P., and P.P. designed research; P.B., T.F., D.H., and J.T.B. performed research; P.B., T.F., D.H., R.P., J.T.B., and P.P. analyzed data; and P.B., T.F., R.P., J.T.B., and P.P. wrote the manuscript.

## Competing interests

The authors declare no competing interests.
