## [Peer Review File · Nature Communications]

Reviewers' Comments:

Reviewer #1:

Remarks to the Author:

Review for NCOMM-19-38620-T

The manuscript "Structure and mechanism of DNA delivery of a gene transfer agent" by Bardy et al reports on the structure of gene transfer agents in *Rhodobacter capsulatus* ("RcGTAs"). Overall this is a fascinating topic in biology and an important one to study as gene transfer agents are linked to pathogenicity and horizontal gene transfer among bacteria. Thus, a mechanistic study of the phage-like particles themselves is warranted and expected to be impactful to a broad audience.

This study relies almost entirely on cryo-EM to decipher the structure and function of these particles. Bardy et al also make use of mass spectrometry and evolutionary relationships among related phage proteins for analysis of the particles. Largely, the high-resolution structures reported in this paper are beautiful (very nice figures as well!) and I have little concern over the technical rigor over the imaging and reconstruction. I do however have some reservations about several conclusions drawn from this work without follow up experimentation. In addition, the presentation of the manuscript needs serious work in order to be suitable for a broad-audience targeted journal. As it is currently written, the results are overly dense and more suited to a heavily structure-based journal.

Specific comments are listed below:

Overall the introduction is well written and clearly states several interesting points about RcGTAs. One question I had is how and why starvation leads to expression of RcGTAs? It seems counterintuitive that this would be the case as it would require a lot of energy from the host to express and assemble the RcGTAs and under starvation conditions there might not be adequate resources available. Can this be explained more?

It is very interesting that the capsid size of the RcGTAs can fit essentially a single *Rhodobacter* gene given the volume. What is known about which genes are selectively packaged? This is an area where deep sequencing of purified particles could reveal a great deal about the biology of the RcGTAs and provide more biological relevance of the structures.

In general, throughout the text several lengthy descriptions of domain boundaries are given and details of individual protein domains are described ad nauseum (example is lines 91-98). The HK97 fold and other phage folds are quite well-established. Technical jargon should be eliminated. One suggestion is to list the residue numbers and domain names in the figures (such as in the color-coded protein schematics) rather than in the text. Reading these detailed descriptions was tedious and dry and did not provide much relevance.

A raw micrograph showing the oblate and isometric heads would be useful to include.

EMDB submission of maps and validation of all models should be required.

The comparison of secondary structure elements between the tape measure protein and the tail needle protein from P22 are not convincing (lines 265-269). This is an odd comparison given the structures of these two types of phage tails. Is there some labeling or other means of identifying this density corresponds to this protein?

Page 9 contains an enormous amount of speculation. I am not convinced there is enough evidence to make reasonable conclusions about the peptidase. Can the authors expand on how they came to these conclusions? Process of elimination? Copy number? Mistakenly identifying density in complex maps is easy to do, so this reviewer would need much more proof of where these proteins reside other than the structure alone. Labels? Knockouts?

The overly large MOI of particles used in the image of Figure 6 is also problematic and makes these data impossible to accurately analyze. What are the statistics of how many phage particles are bound to the cell surface and what is the distribution of orientations? How many images were analyzed? There is nothing I could see in the M&M that describes this experiment.

Reviewer #2:

Remarks to the Author:

The manuscript by Bardy and co-authors "Structure and mechanism of DNA delivery 1 of a gene transfer agent" reports a very interesting and important structural study of a phage-like complex produced by the alphaproteobacterium *Rhodobacter capsulatus*. Some bacteria use phage-like complexes (gene transfer agents or GTAs) to mediate horizontal gene transfer. The genetic exchange using GTA complexes is linked to bacterial adaptation and evolution.

Bardy and co-authors report the first nearly complete structure of the GTA of *Rhodobacter capsulatus* (RcGTA) obtained by cryo-EM. The organization of the RcGTA reminds a tailed double-stranded DNA phage. Each complex is packed with a short piece of DNA (~4,500 bp) from the bacterial genome. Apparently, pieces of genome are selected randomly, and they do not contain information on the GTA structural elements. The authors of this MS describe a number of interesting facts: overall organization of particles has a phage-like appearance, these GTAs have oblate heads, their sizes are lesser compared to the sizes of smallest phages from the subfamily Picovirinae of Podoviridae. The authors have obtained the complete structure of the capsid, in which capsid proteins, portal proteins were identified allowing to reveal points of interactions between them. The authors have determined the structure of the tail which represent some sort a hybrid formation between tails of the Siphoviridae and Podoviridae phages: the particle has a small tail which is longer compared to Podoviridae with a few rings forming the tail tube like in Siphoviridae. Like all phages these phage-like particles have rather characteristic organization of the tail with the tail tip and fibers. The authors have done considerable work in analysis of essential elements of the complex and suggested a hypothesis of its function. This analysis will help to reveal processes of the genome exchange, propagation, infection, and evolution related to the development of the bacterial resistance to medical drugs.

There are some questions that should be addressed by authors to improve the MS.

What is a difference between the head measure and the tape measure proteins (see the first paragraph on page 4)? Why do the authors expect to find a measure protein within the capsid? Typically, dsDNA phages do not have additional proteins within the capsids. So far in other studies related to phages a head measure proteins were not identified. It would be good to have a definition of this protein and what can it define? As it has been shown so far sizes of the heads depend on scaffolding proteins, components involved during self-assembly, and genome size. This first paragraph (page 4) is not consistent with overall information on the phages and results shown in Fig S25 (A and B). Mio- and Siphoviridae families have a separate process for the assemblage of tails, where this tape measure protein became essential. This paragraph is not consistent with middle paragraph in page 7. The conclusion in the paragraph on page 7 (the last sentence) is unjustified.

Head spikes. It would be good if the authors will explain, what do they name as spikes and what is a base protein. Is it located on the outer surface of the capsid? Where are the fibers located? It seems that , they are not resolved?

The authors have to explain themselves: "A dodecamer of portal protein subunits (Rcc01684, g3) replaces one of the two pentamers of major capsid proteins positioned on the fivefold axis of the oblate RcGTA capsid". The sentence is very confusing: although the capsid of the GTA is oblate, it still has 12 5-fold vertices. It is the overall structure that does not have 12 5-fold axis of symmetry, but only one. It is the same for all tailed phages. So which one is occupied by the portal complex?

The authors did not provide an explanation, how they have obtained the structure of the portal protein complex with imposed 12f symmetry and what they have seen in the structure obtained without any symmetry. The indication of interactions between capsids proteins and the portal complex within the capsid where a strong mismatch of symmetries takes place remains unconvincing. To justify changes in symmetry the authors have to provide both figures with the fittings and assessments of differences between different conformations. Possibly, the authors have to find at least three different conformations for the capsid protein or even all five can be different. Possibly the contacts have to break 12f symmetry of the portal complex as well.

Stopper proteins do not regulate the release of the dsDNA, or the authors have to express their ideas a bit clearer. As it has been shown previously by other authors a role of the stopper proteins is to lock the last end of the dsDNA packaged into the procapsid prior attachment of the tail to the phage head. The release of the genome into the bacteria is regulated by signals that come from the distal tail tip towards the head-to-tail connector.

It would be interesting to know, how the authors have defined local symmetries of the tail components. According to publications by other groups (Leiman, Morais, Conway) there are significant differences in symmetries of the tail components from 3f to 6f (including 4 and 5f symmetries). How the symmetries were assessed for the presented structures?

I am not sure that it can be said "Coordination of a putative cation by side chains of Asp210 of major capsid proteins and Glu12 of base proteins strengthens the attachment of head fibers to the RcGTA head." What kind of indications were found for the presence of cations in the structure? It seems that the structure of the head fibers was not well defined. The authors did not present any clear evidence for that.

Format of figures is a bit confusing: even in one figure the same elements are shown at different scales making meaningful comparisons rather difficult.

My impression that the authors have done an important study and found many interesting features of the analysed GTA particles. The study is important, but the authors did not achieve the best resolution that would allow them to trace all proteins de novo and to make strong conclusions on fine features of the complex. It is not an easy job. Unfortunately, the authors did not provide detail information how the polypeptide chains were traced and possibly some parts of them were represented by polyalanines?

The authors have done excellent structural job but done some overfitting and overinterpretation. The logistic of the MS should be improved so the repetitions could be removed, the number of supplementary figures should be reduced, and they should be made more concise to help to understand what it similar and what is special between phage-like particles and picovirinae phages. It

would be important to clarify a biological role and impact of such particles designed by Nature.

It is strongly recommended to do proofreading for English of the MS.

Minor comments

It seems that the authors submitted the MS to another journal, then they have modified it and numbering of figures became confusing: supplementary figures are starting from number 5, and the suppl. figures 1-4 are coming much later. That caused some disruptions in logistic: the topic of the tape measure protein was raised in different parts of the MS Page 4.

Separation of the portal protein of the certain type of domains has been done not in 2015 for the T4 portal protein, but at least in 2007 or even earlier in 2003. Please use the correct reference.

Why is it so important to show some distances between amino acids and not to show them as fitted in the EM maps? The distance were actually not discussed in the text.

Figure 3D -> difficult to understand where are points of interactions between portal, capsid and adaptor proteins; 3E -> the points of interactions are not clear. Was it possible to reveal interactions between certain amino acids?

Response to reviewers' comments to manuscript NCOMMS-19-38620-T "Structure and mechanism of DNA delivery of a gene transfer agent". *Reviewers' comments are highlighted in blue italics*, our **responses in bold black**. Please note that the line numbers localizing changes in the manuscript, which are provided below, refer to the version of manuscript with tracked changes.

Reviewer #1 (Remarks to the Author):

The manuscript "Structure and mechanism of DNA delivery of a gene transfer agent" by Bardy et al reports on the structure of gene transfer agents in Rhodobacter capsulatus ("RcGTAs"). Overall this is a fascinating topic in biology and an important one to study as gene transfer agents are linked to pathogenicity and horizontal gene transfer among bacteria. Thus, a mechanistic study of the phage-like particles themselves is warranted and expected to be impactful to a broad audience.

This study relies almost entirely on cryo-EM to decipher the structure and function of these particles. Bardy et al also make use of mass spectrometry and evolutionary relationships among related phage proteins for analysis of the particles. Largely, the high-resolution structures reported in this paper are beautiful (very nice figures as well!) and I have little concern over the technical rigor over the imaging and reconstruction. I do however have some reservations about several conclusions drawn from this work without follow up experimentation. In addition, the presentation of the manuscript needs serious work in order to be suitable for a broad-audience targeted journal. As it is currently written, the results are overly dense and more suited to a heavily structure-based journal.

Specific comments are listed below:

Overall the introduction is well written and clearly states several interesting points about RcGTAs. One question I had is how and why starvation leads to expression of RcGTAs? It seems counterintuitive that this would be the case as it would require a lot of energy from the host to express and assemble the RcGTAs and under starvation conditions there might not be adequate resources available. Can this be explained more?

A: We have reformulated the sentence and included additional explanation of the reviewers' question: "How and why starvation leads to expression of RcGTAs?" in the introduction section. We also included additional references that address the reviewers' question (lines 41-45):

"The production of the RcGTA is stimulated by nutrient depletion, which induces entry of R. capsulatus into the stationary phase, and high population density detected by quorum sensing(6-8). Similarly, the recipient capability of R. capsulatus cells for RcGTA as well as competence systems of naturally transformable bacteria are highest in the stationary phase, which can be induced by limited availability of nutrients(7, 9, 10)."

It is very interesting that the capsid size of the RcGTAs can fit essentially a single Rhodobacter gene given the volume. What is known about which genes are selectively packaged? This is an area where deep sequencing of purified particles could reveal a great deal about the biology of the RcGTAs and provide more biological relevance of the structures.

A: We have now expanded the introduction to also address this comment by reviewer #1 (lines 51-53):

"RcGTA particles encapsulate all genes of R. capsulatus, however, the genes encoding the proteins forming the RcGTA particle are packaged with a lower

frequency than other regions of the bacterial genome(14). It was speculated that high levels of RcGTA gene transcription in the RcGTA-producing cells cause a reduction in their packaging frequency(14).”

In general, throughout the text several lengthy descriptions of domain boundaries are given and details of individual protein domains are described ad nauseum (example is lines 91-98). The HK97 fold and other phage folds are quite well-established. Technical jargon should be eliminated. One suggestion is to list the residue numbers and domain names in the figures (such as in the color-coded protein schematics) rather than in the text. Reading these detailed descriptions was tedious and dry and did not provide much relevance.

A: As suggested by the reviewer, we have now included residue numbers to indicate the boundaries of domains in figures. We shortened and removed the domain descriptions throughout the manuscript:

We removed the domain description of the major capsid protein from the text – it is now limited to one sentence (lines 146-147):

“The 398-residue-long major capsid protein of RcGTA (Rcc01687, g5) has the canonical HK97 fold shared by tailed phages and herpesviruses(27) (Fig. 2A).”

The description of the portal complex has been shortened by removing the sentence:

“The structure of the portal protein could be built for residues 24-393 out of 396.”

The description of stopper protein has been shortened by removing the sentence:

“The structure of the 112 residue-long stopper protein could be built except for the first two N-terminal residues.”

The description of terminator protein has been shortened by removing the sentence:

“The structure of the 135 residue-long tail terminator protein could be built except for the first N-terminal residue (Fig. 4A).”

The description of tail tube protein has been shortened by removing the sentence:

“The structure of the tail tube protein could be built for residues 3-136 out of 137.”

The description of the hub protein has been shortened by removing the sentence:

“The structure of the hub protein could be built for residues 4-295 out of 296.”

A raw micrograph showing the oblate and isometric heads would be useful to include.

A: Micrographs of particles with oblate and isometric heads have now been included as insets in Fig. 1A and C, respectively.

EMDB submission of maps and validation of all models should be required.

A: EMBD and PDB codes have now been included in Table S1.

The comparison of secondary structure elements between the tape measure protein and the tail needle protein from P22 are not convincing (lines 265-269). This is an odd comparison given the structures of these two types of phage tails. Is there some labeling or other means of identifying this density corresponds to this protein?

A: We have now included statistics describing the sequence and secondary structure prediction similarities of Rcc01694 and g11 of RcGTA to the tail needle protein of P22. In addition, we show the fit of the tail needle protein of P22 in the cryo-EM density of RcGTA to clearly demonstrate that the proteins are structurally similar. (Figure 4B-G and lines 724-729):

“Nevertheless, the content of secondary structure elements and the ability to form rod-shaped trimers of the RcGTA tape measure protein resemble those of tail needle protein gp26 of phage P22 from the family Podoviridae (Fig. 4E-G)(43). Sixty-six residues from the predicted α -helix of g11 of RcGTA can be aligned to residues of gp26 of phage P22 with an e-value of 0.007 and similarity score of 21%, as determined using the program HHpred(44). This indicates that tape measure proteins of long-

tailed phages and tail needle proteins of short-tailed phages may have originated from a common precursor.”

Page 9 contains an enormous amount of speculation. I am not convinced there is enough evidence to make reasonable conclusions about the peptidase. Can the authors expand on how they came to these conclusions? Process of elimination? Copy number? Mistakenly identifying density in complex maps is easy to do, so this reviewer would need much more proof of where these proteins reside other than the structure alone. Labels? Knockouts?

A: We have modified the manuscript to state what leads us to suggest that the tail of RcGTA contains one copy of peptidase. However, we agree with reviewer #1 that we do not provide proof of the nature of the density. We have now shortened the paragraph and removed speculations about the function of the peptidase during the infection process (lines 806-815):

“The central channel of the RcGTA tail contains a fragmented density located between the iris-like constriction formed by the penetration domains of megatron proteins and the C-terminal domains of tape measure proteins (Fig. 1A, S18A). The classification of electron micrographs of RcGTA particles showed that this density is also present in aberrant particles lacking the tape measure protein (Fig. S18AB). The volume of the density corresponds to the molecular mass of a single monomer of the 150-residue-long peptidase (Rcc01697, g14), which was shown to be capable of degrading peptidoglycan from *R. capsulatus* cells(11). Attempts to calculate an asymmetric reconstruction of the peptidase were unsuccessful, probably due to its small mass. The deletion of the peptidase gene prevented the formation of native particles of RcGTA(11), and we observed empty heads without tails (Fig. S19). Therefore, it cannot be determined with certainty that the unassigned density in the RcGTA tail belongs to RcGTA peptidase g14.”

The overly large MOI of particles used in the image of Figure 6 is also problematic and makes these data impossible to accurately analyze. What are the statistics of how many phage particles are bound to the cell surface and what is the distribution of orientations? How many images were analyzed? There is nothing I could see in the M&M that describes this experiment.

A: We have now included additional figures and analysis of the directions of the tails of RcGTA (Fig. 6A-E). We have now modified the manuscript to explain the variations in numbers of RcGTA particles attached to different cells (lines 890-892):

“Cells of *R. capsulatus* are heterogeneous in their capacity to bind RcGTA, since numerous RcGTA particles bind to some cells, whereas others do not attract any (Fig. 6C-E).”

Furthermore, we have now included an extra chapter in the Supplementary Materials and Methods section to explain details of the experiment and provide analysis of the directions of RcGTA tails (lines 452-465):

“Quantification of attachment of RcGTA to *R. capsulatus* cells

R. capsulatus strain B10 was grown overnight in RCV medium to OD_{600nm} of 0.5 to 1.0 (29). The cells were harvested by centrifugation at 3000 g for 10 min, the supernatant was discarded and the pellet was resuspended in the same volume of G-buffer. This step was repeated 4 times. The pellet from the last step was resuspended in G-buffer to obtain a final OD_{600nm} of 32. DNase was added to the sample to a final concentration of 1 μ g/ml, and the sample was incubated for 30 min at room temperature. Subsequently, 4 μ l of cell suspension was mixed with 1 μ l of purified RcGTA particles in G-buffer at A_{280nm} = 1.5 (corresponding to an MOI of Rif-transferring GTA of 0.0002, measured using a gene transferring assay (46)) and incubated for additional 5-10 minutes. A sample volume of 3.9 μ l was then applied onto glow-discharged holey carbon-coated copper grids (R2/1 200 mesh, Quantifoil),

blotted and plunge-frozen and analyzed in an electron microscope operated at 200 kV. For the statistical evaluation of tail orientations, seven micrographs were analyzed from each grid of a biological triplicate (21 micrographs in total), obtaining data from 74 empty and 125 native particles.”

Reviewer #2 (Remarks to the Author):

The manuscript by Bardy and co-authors “Structure and mechanism of DNA delivery of a gene transfer agent” reports a very interesting and important structural study of a phage-like complex produced by the alphaproteobacterium Rhodobacter capsulatus. Some bacteria use phage-like complexes (gene transfer agents or GTAs) to mediate horizontal gene transfer. The genetic exchange using GTA complexes is linked to bacterial adaptation and evolution.

Bardy and co-authors report the first nearly complete structure of the GTA of Rhodobacter capsulatus (RcGTA) obtained by cryo-EM. The organization of the RcGTA reminds a tailed double-stranded DNA phage. Each complex is packed with a short piece of DNA (~4,500 kb pairs) from the bacterial genome. Apparently, pieces of genome are selected randomly, and they do not contain information on the GTA structural elements. The authors of this MS describe a number of interesting facts: overall organization of particles has a phage-like appearance, these GTAs have oblate heads, their sizes are lesser compared to the sizes of smallest phages from the subfamily Picovirinae of Podoviridae. The authors have obtained the complete structure of the capsid, in which capsid proteins, portal proteins were identified allowing to reveal points of interactions between them. The authors have determined the structure of the tail which represent some sort a hybrid formation between tails of the Siphoviridae and Podoviridae phages: the particle has a small tail which is longer compared to Podoviridae with a few rings forming the tail tube like in Siphoviridae. Like all phages these phage-like particles have rather characteristic organization of the tail with the tail tip and fibers.

The authors have done considerable work in analysis of essential elements of the complex and suggested a hypothesis of its function. This analysis will help to reveal processes of the genome exchange, propagation, infection, and evolution related to the development of the bacterial resistance to medical drugs.

There are some questions that should be addressed by authors to improve the MS.

What is a difference between the head measure and the tape measure proteins (see the first paragraph on page 4)? Why do the authors expect to find a measure protein within the capsid? Typically, dsDNA phages do not have additional proteins within the capsids. So far in other studies related to phages a head measure proteins were not identified. It would be good to have a definition of this protein and what can it define? As it has been shown so far sizes of the heads depend on scaffolding proteins, components involved during self-assembly, and genome size. This first paragraph (page 4) is not consistent with overall information on the phages and results shown in Fig S25 (A and B). Mio- and Siphoviridae families have a separate process for the assemblage of tails, where this tape measure protein became essential. This paragraph is not consistent with middle paragraph in page 7.

A: We wish to thank reviewer #2 for this comment. Our use of the term “tape measure protein” in the first paragraph on page 4 was misleading. We have now rewritten the first paragraph on page 4 to omit the term “tape measure protein” (lines 220-227): “The existence of RcGTA particles with icosahedral heads provides evidence that the assembly of the oblate capsid is not based on intrinsic properties of the RcGTA major capsid protein but may be determined by scaffolding proteins. The locus of genes encoding proteins forming the head of RcGTA includes a hypothetical Rcc01685 with an as-yet unknown function (Fig. 1B, S7A). Rcc01685 has the predicted structure of a 75-residue-long α -helix, similar to that of the scaffolding protein of phage phi29 from the family Podoviridae (Fig. S7B-D)(28). However, an R. capsulatus knock-out of

Rcc01685(14) produces oblate capsids (Fig. S7E). Therefore, other proteins must be responsible for determining the head shape of RcGTA.”

The conclusion in the middle paragraph on page 7 (the last sentence) is unjustified.

A: We have now removed the speculation from the middle paragraph on page 7 and from the abstract as well:

Text (lines 757-758): “This indicates that tape measure proteins of long-tailed phages and tail needle proteins of short-tailed phages may have originated from a common precursor.”

Abstract (lines 23-25): “The tail channel of RcGTA contains a trimer of proteins that possess features of both tape measure proteins of long-tailed phages from the family Siphoviridae and tail needle proteins of short-tailed phages from the family Podoviridae.”

Head spikes. It would be good if the authors will explain, what do they name as spikes and what is a base protein. Is it located on the outer surface of the capsid? Where are the fibers located? It seems that, they are not resolved?

A: We have now rewritten the text describing the structure of head spikes to provide a clear explanation of what we mean by head spikes, base proteins, and head fibers (lines 229-234):

“The surface of the RcGTA head is decorated with eleven 70-Å-long head spikes attached to pentamers of major capsid proteins (Fig. 1A, S1-3, Table S1). Each head spike is composed of a pentamer of base proteins Rcc01079, and a single subunit of head fiber protein Rcc01080 (Fig. 2B-F). The base protein has a jellyroll fold formed by β -strands 1-6. Proteins of similar fold form protrusions at the surfaces of bacterial, archaeal, and eukaryotic viruses (Table S4). Base proteins attach to the capsid by the N-termini, each of which binds to axial domains of two adjacent major capsid proteins within a pentamer (Fig. 2B-D).”

The authors have to explain themselves: “A dodecamer of portal protein subunits (Rcc01684, g3) replaces one of the two pentamers of major capsid proteins positioned on the fivefold axis of the oblate RcGTA capsid”. The sentence is very confusing: although the capsid of the GTA is oblate, it still has 12 5-fold vertices. It is the overall structure that does not have 12 5-fold axis of symmetry, but only one. It is the same for all tailed phages. So which one is occupied by the portal complex?

A: We have now modified the sentence to make it clearer: (lines 244-245):

“The RcGTA head contains a special vertex in which a dodecamer of portal protein subunits (Rcc01684, g3) replaces a pentamer of major capsid proteins (Fig. 1A, 3A).”

The authors did not provide an explanation, how they have obtained the structure of the portal protein complex with imposed 12f symmetry and what they have seen in the structure obtained without any symmetry.

A: The sequence of steps describing the structure determination of RcGTA particle is described in Supplementary Material and Methods (Supplementary material - lines 371-384):

“Reconstruction of portal/neck region

The reconstruction of the portal and neck region was initiated by a refinement with a full search of rotation angle and local searches of tilt and psi angles on twice-binned images. Subsequently, three-dimensional classification and final refinement were performed using unbinned data. Because of the symmetry mismatch between the

capsid and portal/neck region, a mask excluding the capsid was applied. For the reconstruction of the portal and adaptor complexes with C12 symmetry, the neck with C6 symmetry was also masked out. Three-dimensional classification with the skip_align option and C6 symmetry, based on the orientations from the C12 reconstruction, was performed to obtain two separate reconstructions of the tail rotated 30° relative to each other. Orientations of particles from one of the groups were adjusted by the addition of 30° to their rotation angle. Where the adjusted value of the rotation angle exceeded 30°, we subtracted 60° from its value to restrict the rotation angle values to the range of -30° to +30° of C6 symmetry. The reconstruction continued with refinement with local searches on the twice-binned data, three-dimensional classification and the final refinement of unbinned data.”

Description of the asymmetric structure of the portal complex has now been included in the manuscript and is shown in Fig. S9 (lines 259-261):

“The asymmetric reconstruction of the RcGTA particle shows that interactions of the capsid with portal and adaptor complexes induce deviations of the structures from their ideal fivefold and twelvefold symmetries, respectively (Fig. S9).”

The indication of interactions between capsids proteins and the portal complex within the capsid where a strong mismatch of symmetries takes place remains unconvincing. To justify changes in symmetry the authors have to provide both figures with the fittings and assessments of differences between different conformations. Possibly, the authors have to find at least three different conformations for the capsid protein or even all five can be different. Possibly the contacts have to break 12f symmetry of the portal complex as well.

A: We have now modified Fig. 3B-E to clearly show the structural differences between the subunits of capsid, portal, and adaptor proteins. Furthermore, we have now included a new Fig. S9 that characterizes structural differences between subunits of the complexes in different conformations. This comparison clearly demonstrates that the structures are indeed asymmetric. We have now included an additional sentence in the manuscript to highlight the structural differences in the asymmetric reconstruction of the neck region of the RcGTA particle (lines 259-261):

“The asymmetric reconstruction of the RcGTA particle shows that interactions of the capsid with portal and adaptor complexes induce deviations of the structures from their ideal fivefold and twelvefold symmetries, respectively (Fig. S9).”

Stopper proteins do not regulate the release of the dsDNA, or the authors have to express their ideas a bit clearer. As it has been shown previously by other authors a role of the stopper proteins is to lock the last end of the dsDNA packaged into the procapsid prior attachment of the tail to the phage head. The release of the genome into the bacteria is regulated by signals that come from the distal tail tip towards the heat-to-tail connector.

A: We have now rewritten the paragraph describing the structure and putative function of RcGTA stopper proteins (lines 605-614):

“A hexamer of RcGTA stopper proteins (Rcc01689, g7) binds to the adaptor complex (Fig. 1A, 4A, S10AB, Table S5). Each 112-residue-long stopper protein interacts with adaptor loops from two neighboring adaptor proteins (Fig. S10AB). At the distal interface, the long and insertion loops and C-terminus of the stopper protein bind to a terminator protein (Fig. 4A, S10CD). The stopper protein of RcGTA is named after its homologue gp16 from bacteriophage SPP1 (Fig. 4C). It has been speculated that the long loop of gp16 holds the genome of SPP1 inside the phage head and regulates its release(39). However, the long loop of the RcGTA stopper protein interacts with the tail terminator protein and does not block the central tail channel (Fig. 4AC).

Furthermore, the end of the DNA packaged in the native RcGTA particle extends through the channel formed by the hexamer of stopper proteins and continues into the disc of tail terminator proteins (Fig. 1A, 3G).”

It would be interesting to know, how the authors have defined local symmetries of the tail components. According to publications by other groups (Leiman, Morais, Conway) there are significant differences in symmetries of the tail components from 3f to 6f (including 4 and 5f symmetries). How the symmetries were assessed for the presented structures?

A: In the process of structure determination, we calculated the structures of various parts of the RcGTA tail with different symmetries, including C1. The symmetry of various complexes was determined based on the interpretability of the resulting maps. Structures of RcGTA components reached resolutions that enabled us to build atomic models, therefore we are reasonably certain that our assignment of symmetry is correct.

I am not sure that it can be said “Coordination of a putative cation by side chains of Asp210 of major capsid proteins and Glu12 of base proteins strengthens the attachment of head fibers to the RcGTA head.” What kind of indications were found for the presence of cations in the structure? It seems that the structure of the head fibers was not well defined. The authors did not present any clear evidence for that.

A: Thank you, the sentence was indeed incorrect. It should read (lines 955-957): “Coordination of a putative cation by side chains of Asp210 of major capsid proteins and Glu12 of base proteins strengthens the attachment of base proteins to the RcGTA head.”

Format of figures is a bit confusing: even in one figure the same elements are shown at different scales making meaningful comparisons rather difficult.

A: We have now modified panels B and E of Fig. 2 to be on the same scale.

My impression that the authors have done an important study and found many interesting features of the analysed GTA particles. The study is important, but the authors did not achieve the best resolution that would allow them to trace all proteins de novo and to make strong conclusions on fine features of the complex. It is not an easy job. Unfortunately, the authors did not provide detail information how the polypeptide chains were traced and possibly some parts of them were represented by polyalanines?

A: Information on how the individual protein structures of RcGTA particle were built has now been included in Table S2. The table also contains information on which parts of proteins were modelled as poly-alanine chains because of insufficient resolution of the available cryo-EM maps is included in the PDB validation reports.

The authors have done excellent structural job but done some overfitting and overinterpretation. The logistic of the MS should be improved so the repetitions could be removed, the number of supplementary figures should be reduced, and they should be made more concise to help to understand what is similar and what is special between phage-like particles and picovirinae phages. It would be important to clarify a biological role and impact of such particles designed by Nature.

A: We have now shortened the manuscript according to the recommendations of both reviewers. Furthermore, we included extra text into the Introduction to highlight the importance of GTAs in gene transfer among marine bacteria. We have reduced the number of supplementary figures by 7. We believe that the changes have made the manuscript concise and more readable.

It is strongly recommended to do proofreading for English of the MS.

A: The manuscript has been checked by a professional English proof-reading service.

Minor comments

It seems that the authors submitted the MS to another journal, then they have modified it and numbering of figures became confusing: supplementary figures are starting from number 5, and the suppl. figures 1-4 are coming much later. That caused some disruptions in logistic: the topic of the tape measure protein was raised in different parts of the MS Page 4.

A: Thank you, we have now corrected the numbering of supplementary figures.

Separation of the portal protein of the certain type of domains has been done not in 2015 for the T4 portal protein, but at least in 2007 or even earlier in 2003. Please use the correct reference.

A: We have now included reference to the structure of the portal protein from the year 2007:

A. A. Lebedev et al., Structural framework for DNA translocation via the viral portal protein. EMBO J 26, 1984-1994 (2007).

Why is it so important to show some distances between amino acids and not to show them as fitted in the EM maps? The distance were actually not discussed in the text.

A: We did not include the EM maps in most of the main figures because they detracted from the clarity of the display. Distances between selected amino acids are provided as an indication of their interactions.

Figure 3D -> difficult to understand where are points of interactions between portal, capsid and adaptor proteins; 3E -> the points of interactions are not clear. Was it possible to reveal interactions between certain amino acids?

A: We have now modified the figures to clearly highlight differences between the protein structures. In addition, we have now included Fig. S9 that provides a real space correlation coefficient analysis to show that the subunits of the capsid, portal, and adaptor proteins in the neck region have unique structures because of the asymmetric interactions.

Reviewers' Comments:

Reviewer #1:

Remarks to the Author:

Thank you to the authors for carefully addressing the points raised during my initial review. With removed speculation and cleaner, more streamlined language the new manuscript is much improved. Congratulations on such nice work!

Reviewer #2:

Remarks to the Author:

The MS by Bardy and co-authors was significantly improved and much easier to read and follow the logic of the results.

There are only very minor comments

Lines 25-27 -> abstract

"The opening of a constriction within the RcGTA baseplate is mechanically linked to the exposure of membrane-penetration helices at the tip of the tail and enables the ejection of DNA into the periplasm of recipient bacteria." It should be made clear, that this is a hypothesis, since the resolution of the distal end of the tail was not brilliant (~ 15Å?).

Lines 102-104

"Additional differences among the capsid proteins are in residues 217-222 from the axial domain, which form a loop in a hexamer, but are disordered in the major capsid proteins that constitute pentamers". Apparently the authors talking of the A-loops, but in all conformers they look exactly the same -> figure Fig. S6C (like in hk97).

Would be useful to show this loop in the figure S6C (related to pentamers) -> in this figure in atomic models all A-loops are rather well defined, but possibly not in the map? So it is difficult to understand which loops are disordered.

Lines 230-232

"The structure of tail tube proteins does not change after DNA ejection (Fig. S11), providing evidence that conformational changes of the tail do not propagate a signal about baseplate binding to a cell to trigger DNA ejection." It would be recommended to rephrase the sentence. The authors did not observe changes in the tail after DNA ejection. There are a few reasons: it may be due to the fact, that the tail is short; the authors should be aware that some conformational changes are reversible, so after DNA ejection proteins may restore their initial stable conformation, especially in situ conditions. So far no changes were found in podoviridae phages, at least nothing was reported. The hypothesis of signalling by tails in all types of phage should still be verified, although such signalling really exists in myoviridae phages.

Possibly it would be better to rephrase the sentence in the following way: "The structure of tail tube did not reveal any changes after DNA ejection (Fig. S11), suggesting that the tail is not involved into signalling to trigger DNA ejection after attachment to a cell."

Lines 321-332

Conformational changes of RcGTA baseplate regulate DNA release. -> It seems that this paragraph is too speculative: a resolution of this part of the structure was rather low. So conformational changes were actually not observed. Possibly it would be more safe to avoid unnecessary suggestions, which were not supported by real observations. Projections were not very convincing,

Response to reviewers' comments to manuscript NCOMMS-19-38620A "Structure and mechanism of DNA delivery of a gene transfer agent". *Reviewers' comments are highlighted in blue italics*, our **responses in bold black**. Please note that the line numbers localizing changes in the manuscript, which are provided below, refer to the version of manuscript with tracked changes.

Reviewer #1 (Remarks to the Author):

Thank you to the authors for carefully addressing the points raised during my initial review. With removed speculation and cleaner, more streamlined language the new manuscript is much improved. Congratulations on such nice work!

A: Thank you.

Reviewer #2 (Remarks to the Author):

The MS by Bardy and co-authors was significantly improved and much easier to read and follow the logic of the results.

There are only very minor comments

Lines 25-27 -> abstract

"The opening of a constriction within the RcGTA baseplate is mechanically linked to the exposure of membrane-penetration helices at the tip of the tail and enables the ejection of DNA into the periplasm of recipient bacteria." It should be made clear, that this is a hypothesis, since the resolution of the distal end of the tail was not brilliant (~ 15Å?).

A: We have now shortened the sentence, to remove the speculative part. This modification also shortened the abstract to fit within the 150 words limit. Lines 25-26: "The opening of a constriction within the RcGTA baseplate enables the ejection of DNA into bacterial periplasm."

Lines 102-104

"Additional differences among the capsid proteins are in residues 217-222 from the axial domain, which form a loop in a hexamer, but are disordered in the major capsid proteins that constitute pentamers". Apparently the authors talking of the A-loops, but in all conformers they look exactly the same -> figure Fig. S6C (like in hk97).

Would be useful to show this loop in the figure S6C (related to pentamers) -> in this figure in atomic models all A-loops are rather well defined, but possibly not in the map? So it is difficult to understand which loops are disordered.

A: Differences in A-loops of major capsid proteins are a minor point in the manuscript. Modification of the figure as suggested by the reviewer would make the figure difficult to understand. Therefore, we decided to remove the mention of the A-loop differences from the main text. Lines 109-115.

Lines 230-232

"The structure of tail tube proteins does not change after DNA ejection (Fig. S11), providing evidence that conformational changes of the tail do not propagate a signal about baseplate binding to a cell to trigger DNA ejection." It would be recommended to rephrase the sentence. The authors did not observe changes in the tail after DNA ejection. There are a few reasons: it may be due to the fact, that the tail is short; the authors should be aware that some conformational changes are reversible, so after DNA ejection proteins may restore their initial stable conformation, especially in situ conditions. So far no changes were found in podoviridae phages, at least nothing was reported. The hypothesis of signalling by tails in all types of phage should still be verified, although such signalling really exists in myoviridae phages. Possibly it would be better to rephrase the sentence in the following way: "The

structure of tail tube did not reveal any changes after DNA ejection (Fig. S11), suggesting that the tail is not involved into signalling to trigger DNA ejection after attachment to a cell.”

A: We accepted suggestion by the reviewer and modified the manuscript accordingly. Lines 297-299:

“The structure of tail tube did not reveal any changes after DNA ejection (Supplementary Figure 11), suggesting that the tail is not involved in signaling to trigger DNA ejection after attachment to a cell.”

Lines 321-332

Conformational changes of RcGTA baseplate regulate DNA release. -> It seems that this paragraph is too speculative: a resolution of this part of the structure was rather low. So conformational changes were actually not observed. Possibly it would be more safe to avoid unnecessary suggestions, which were not supported by real observations. Projections were not very convincing,

A: We have now removed all references to conformational changes from the entire section to make it less speculative. Lines 434-445:

“Reorganization of baseplate regulates DNA release

The tail channel of the RcGTA native particle is constricted by an iris made from α -helices 1 of megatron proteins (Fig. 5CD). The iris has to open to enable genome ejection. The native baseplate of RcGTA does not contain the space necessary to shift the α -helices 1 away from the pore (Fig. 5ACD). Therefore, the release of DNA from the RcGTA particle requires re-arrangement of the baseplate, which is consistent with the observation that one third of empty RcGTA particles lacked baseplates (Supplementary Figure 18B). The central channel of the RcGTA tail above the iris contains a trimer of tape measure proteins and perhaps also one molecule of peptidoglycan peptidase (Fig. 1A, Supplementary Figure 13EF). These proteins have to be released from the virion before the DNA can exit. Some of the empty particles with the baseplate still present contained density corresponding to the inner tail proteins, which would not allow the DNA ejection through the tail (Supplementary Figure 18B). We speculate that the empty particles with attached baseplate are defective and never contained a full complement of DNA.”